# Translational repression by 4E-T is crucial to maintain the prophase-I arrest in vertebrate oocytes

Andreas Heim [1] ✉, Shiya Cheng [2], Jan Orth[1,3], Florian Stengel [1,3], Melina Schuh [2,4] & Thomas U. Mayer [1,3] ✉

Meiotic maturation of vertebrate oocytes occurs in the near-absence of transcription. Thus, female fertility relies on timely translational activation of maternal transcripts stockpiled in full-grown prophase-I-arrested oocytes. However, how expression of these mRNAs is suppressed to maintain the long-lasting prophase-I arrest remains mysterious. Utilizing fast-acting TRIM-Away, we demonstrate that acute loss of the translation repressor 4E-T triggers spontaneous release from prophase-I arrest in mouse and frog oocytes. This is due to untimely expression of key meiotic drivers like c-Mos and cyclin-B1. Notably, mutant 4E-T associated with premature ovarian insufficiency in women fails to maintain the prophase-I arrest in Xenopus oocytes. We further show that 4E-T association with eIF4E and PATL2 is critical for target mRNA binding and repression. Thus, 4E-T is a central factor in translational repression of mRNAs stockpiled in full-grown oocytes for later activation and, therefore, essential to sustain the oocyte pool throughout the reproductive lifespan of female vertebrates.

In vertebrates, immature oocytes arrest in a G2-like state during MI (referred to as stage-VI in *Xenopus laevis*) characterized by low activity of Cyclin-dependent kinase 1 (Cdk1). This prophase-I arrest has to be highly robust—it can last up to decades in humans—but at the same time also highly responsive to hormonal stimulation triggering meiotic resumption by activating Cdk1[1]. Importantly, meiotic maturation occurs in the near-absence of transcription, so fertility relies on large amounts of maternal mRNAs stockpiled in prophase-I-arrested oocytes[2]. To ensure that the oocytes' proteome matches the specific cellular requirements, regulatory sequences in mRNAs recruit RNA-binding proteins to form ribonucleoparticles (RNP), which control timing and strength of translational activation[3,4]. Cap-dependent translation is initiated by binding of the 43S preinitiation complex (PIC) to an mRNA near its 5′end, which is then scanned for a start codon to trigger assembly of the full 80S ribosome. Recruitment of the 43S PIC is mediated by the eIF4F complex consisting of the cap-binding protein eIF4E, the helicase eIF4A and the scaffold protein eIF4G[5]. eIF4G serves as a platform for interactions with the 43S PIC as well as with the 5′cap via eIF4E and the 3′ polyA-tail via polyA-binding proteins (PABP), thus establishing the so-called "closed-loop" configuration[6–8]. Disrupting the interaction between eIF4E and eIF4G by other eIF4E-binding proteins (4E-BP) is a common mechanism inhibiting translation. Previously, the 4E-BP Maskin was reported to suppress translation of mRNAs bound by the cytoplasmic poly-adenylation element binding protein 1 (CPEB1) in prophase-I-arrested Xenopus oocytes[9]. However, others could not confirm this interaction leaving Maskin's contribution to translation repression in oocytes unanswered[10,11]. Interestingly, another 4E-BP, eIF4E-Transporter (4E-T/EIF4ENIF1), was implicated in binding CPEB1 during early and late oogenesis in Xenopus[10–12] and to be present in the mRNA storage

[1]Department of Biology, University of Konstanz, Konstanz, Germany. [2]Department of Meiosis, Max Planck Institute for Multidisciplinary Sciences, Göttingen, Germany. [3]Konstanz Research School Chemical Biology, University of Konstanz, Konstanz, Germany. [4]Cluster of Excellence "Multiscale Bioimaging: From Molecular Machines to Networks of Excitable Cells" (MBExC), University of Göttingen, Göttingen, Germany. ✉e-mail: Andreas.Heim@uni-konstanz.de; Thomas.U.Mayer@uni-konstanz.de

compartment MARDO in prophase-I-arrested mammalian oocytes[13]. Furthermore, heterozygous 4E-T mutations are associated with premature ovarian insufficiency (POI) in women[14–17]. Nevertheless, the physiological relevance of 4E-T, or in fact any other translation regulator, for the maintenance of the long-lasting prophase-I arrest of full-grown vertebrate oocytes is still largely unknown. This knowledge gap stems, in part, from technical limitations associated with the approaches applied so far, e.g., knock-out or RNA-interference approaches, all of which lack sufficient temporal resolution to precisely dissect the function of a protein at a distinct developmental stage, without interfering with its preceding function.

In this study, we acutely deplete 4E-T from full-grown prophase-I-arrested frog and mouse oocytes without interfering with its function in early oogenesis by applying TRIM-Away[18]. Strikingly, we demonstrate that upon acute loss of 4E-T, mouse and frog oocytes fail to maintain the prophase-I arrest. We further delineate 4E-T's molecular interaction network and contributions of the different interactors to translational repression in oocytes. We identify key target mRNAs encoding M-phase promoting regulators that are under direct control by 4E-T and provide a molecular explanation why women with a heterozygous 4E-T mutation suffer from POI. Therefore, we directly show that continuous translation repression by a 4E-BP is required to maintain the long-lasting prophase-I arrest in full-grown vertebrate oocytes. Mass spectrometry (MS) analyses identified additional proteins with diverse cellular functions to be upregulated in 4E-T depleted oocytes. By integrating our and past data, we develop a comprehensive model on the function and regulation of 4E-T in vertebrate oocytes.

## Results

### 4E-T depletion triggers spontaneous meiotic resumption

Several translation regulators are reported to affect mRNA translation during vertebrate oogenesis[2,19]. However, their relevance for the maintenance of the prophase-I arrest in full-grown oocytes is completely unclear. Since 4E-T has been implicated in translation repression during early oogenesis in *Xenopus laevis* and mice[10,18], we speculated that 4E-T might be critical for the prophase-I arrest in vertebrate oocytes. To test this, we depleted 4E-T from intact oocytes using TRIM-Away, which employs the E3 ligase TRIM21 to degrade endogenous proteins that are bound to an antibody[20–22]. Using distinct peptides, we raised two different antibodies against Xenopus 4E-T (4E-T^Ab1 and 4E-T^Ab2, Fig. S1a) and injected each of them together with Flag-TRIM21 mRNA into stage-VI Xenopus oocytes. Western blot (WB) analyses confirmed efficient depletion of endogenous 4E-T in oocytes injected with 4E-T^Ab1 or 4E-T^Ab2, but not with unspecific (Ctrl) antibody (Fig. 1a). As 4E-T^Ab1 was slightly more efficient in 4E-T depletion, we used this antibody for further studies. Notably, within 35 h after 4E-T^Ab1 injection, nearly all oocytes underwent spontaneous meiotic resumption, determined by the appearance of a white spot in the oocytes' animal hemisphere as readout for GVBD (Fig. 1b). In contrast, almost all Ctrl-depleted oocytes maintained the prophase-I arrest. This phenotype was specific because expression of ectopic 4E-T partially lacking the antigen region (4E-T^TRIM) and, therefore, not recognized by 4E-T^Ab1 (Fig. S1b) rescued spontaneous, PG-independent meiotic resumption (Fig. 1b). To investigate if this function of 4E-T is conserved, we depleted 4E-T from full-grown GV stage mouse oocytes using TRIM-Away (Fig. 1c). In the presence of dibutyryl cyclic AMP (dbcAMP), a non-hydrolysable cAMP analog, none of the Ctrl-depleted oocytes resumed meiosis (Fig. 1d, e). In contrast, all 4E-T-depleted oocytes spontaneously resumed meiosis in the presence of dbcAMP. Around half of these oocytes extruded the first polar body within 9 h after meiotic resumption, a rate comparable to the one of Ctrl-depleted oocytes induced to resume meiosis by dbcAMP withdrawal. Thus, these experiments demonstrate that 4E-T is a translation regulator that is essential for the maintenance of the long-lasting prophase-I arrest in full-grown vertebrate oocytes. Next, we established a more

controllable assay to quantify meiotic maturation by depleting 4E-T for just 22 h from Xenopus oocytes followed by PG stimulation. As expected, compared to Ctrl-depleted oocytes, 4E-T-depleted oocytes using 4E-T^Ab1 reached GVBD much faster, and this was again fully rescued by Flag-4E-T^TRIM (Fig. 1f, g). Depleting 4E-T using 4E-T^Ab2 similarly accelerated meiotic timing (Fig. S1e). Consequentially, inhibitory phosphorylation of Cdk1 disappeared several hours earlier upon 4E-T depletion and this correlated with CPEB1 degradation, a well-known Cdk1-mediated event[23] (Fig. 1h). Depletion of 4E-T from the oocytes led to a striking increase in the expression levels of the Cdk1 activator cyclin-B1, which was suppressed to a large extent by expression of ectopic 4E-T. c-Mos expression and activation of its downstream target MAPK were also highly accelerated and, importantly, these effects could be rescued again. Of note, under rescue conditions, activation of MAPK slightly preceded strong upregulation of c-Mos. We propose that this weak early activation of the MAPK pathway was caused by the fact that suppression of the c-Mos mRNA was not perfectly rescued in this experiment resulting in a small increase in c-Mos protein levels not detectable by immunoblot.

Previously, a heterozygous nonsense mutation in 4E-T (serine429→stop codon) has been genetically correlated with a case of dominantly inherited POI in humans[17]. Importantly, the corresponding truncation of Xenopus 4E-T^TRIM (4E-T^TRIM 1-319) was indeed completely deficient in rescuing GVBD timing after PG treatment of 4E-T-depleted Xenopus oocytes (Fig. 1i, j). This suggests that POI in women with mutated 4E-T is caused by a failure to maintain the long-lasting prophase-I arrest in immature oocytes. Since POI occurred in heterozygous patients, this defect can apparently not be compensated by the non-mutated allele, which is supported by recent studies using haploinsufficient 4E-T knock-out mice[24]. From these data, we concluded that the translation repressor 4E-T has an evolutionarily conserved essential function in maintaining the prophase-I arrest in full-grown oocytes.

### 4E-T controls the translation of cell cycle mRNAs

Next, we investigated why 4E-T-depleted oocytes failed to maintain the prophase-I arrest. We assumed that loss of 4E-T liberates stockpiled mRNAs in stage-VI oocytes from repression causing a detectable increase in overall translational activity. To test this, we incubated stage-VI oocytes with puromycin, whose rate of incorporation into nascent polypeptides can be used to estimate protein synthesis rate[25]. Importantly, global protein synthesis was strongly upregulated upon 4E-T depletion, and this could be largely rescued by Flag-4E-T^TRIM (Fig. 2a). The incomplete rescue of the native translational state could be related to the partial co-depletion of some proteins that strongly interact with 4E-T (Fig. S1c, d). Such a phenomenon can be frequently observed when single components of protein complexes are experimentally destabilized[26]. Thus, although re-expression of just 4E-T was sufficient to fully restore the ability of oocytes to maintain the prophase-I arrest (Fig. 1b), it might not have been sufficient to completely restore the native translational state of mRNAs. Next, we speculated that the strong and robust increase in translation induced by 4E-T depletion involves key meiotic regulators. To test this, we used either Ctrl or 4E-T antibodies for immunoprecipitation (IP) from lysates of Xenopus stage-VI oocytes, isolated associated RNAs and performed qRT-PCR against mRNAs encoding c-Mos, cyclin-B1 (CCNB1), cyclin-A1 (CCNA1), TPX2, BTG4, Wee2 and XErp1, all factors upregulated during meiotic maturation[27–36]. Intriguingly, all mRNAs were strongly enriched in the 4E-T IP compared to the Ctrl IP, while control mRNAs encoding the globins HBG1 and HBG2 or the actin subunit ACTA2 were not significantly enriched (Fig. 2b). These data demonstrate that 4E-T binds to mRNAs encoding key M-phase promoting factors during the prophase-I arrest. To test if 4E-T, bound to these mRNAs, represses their translation, we performed reporter assays using the 3′UTR of c-Mos, CCNB1 and CCNA1 mRNAs fused to the open reading frame (ORF) of Flag-eGFP. The 3′UTR of the HBG1

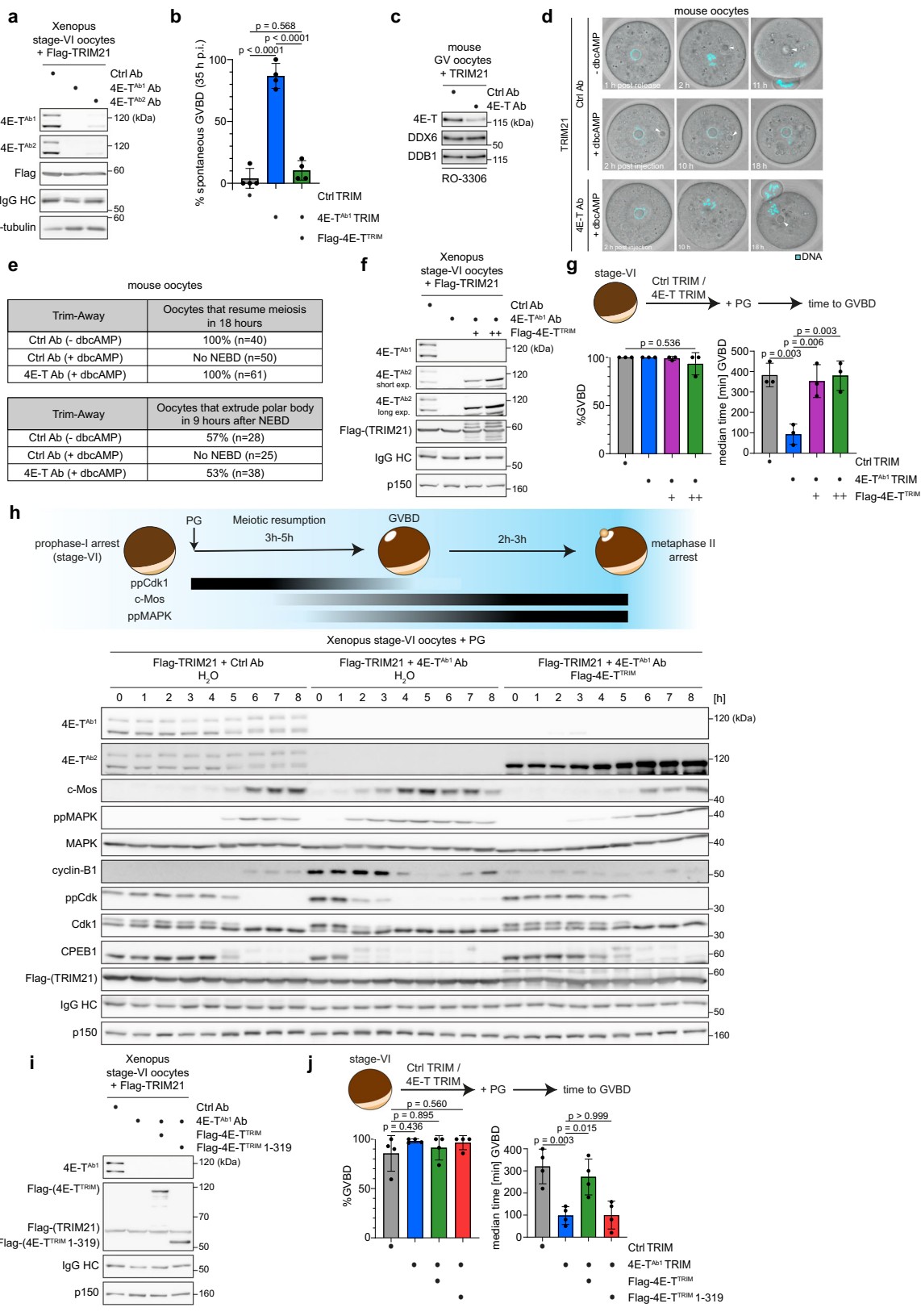

mRNA fused to Myc-eGFP served as control. Each reporter mRNA was co-injected with the HBG1 control reporter, TRIM21 mRNA and Ctrl or 4E-T Ab into Xenopus stage-VI oocytes. Importantly, depletion of 4E-T was sufficient to upregulate expression of the c-Mos and cyclin-B1 reporter mRNAs, and this was fully rescued by 4E-T[TRIM] (Fig. 2c, d). As expected, translation of the HBG1 reporter mRNA was not

upregulated. Surprisingly, although the cyclin-A1 mRNA was strongly associated with 4E-T (Fig. 2b), CCNA1 reporter mRNA expression was not significantly affected by 4E-T depletion, suggesting that additional regulatory layers exist. From these results, we predicted that 4E-T depletion should result in a detectable increase in endogenous c-Mos and cyclin-B1 protein levels in prophase-I-arrested oocytes. To test this,

**Fig. 1 | 4E-T depletion triggers spontaneous meiotic resumption and accelerates hormone-induced meiotic maturation. a** Xenopus stage-VI oocytes were injected with the indicated 4E-T or unspecific control (Ctrl) antibodies and mRNA encoding Flag-TRIM21. Samples were taken 22 h after injection and immunoblotted as indicated. One representative experiment of three independent biological replicates is shown. **b** Xenopus stage-VI oocytes were injected with water or mRNA encoding Flag-4E-T^TRIM. 18 h after injection, oocytes were co-injected with mRNA encoding Flag-TRIM21 and either 4E-T^Abl or unspecific control (Ctrl) antibodies. 35 h after the second injection, the occurrence of GVBD was determined by the appearance of a white spot in the animal hemisphere of the oocytes (Ctrl TRIM + H$_2$O, $n = 107$ oocytes; 4E-T^Abl TRIM + H$_2$O, $n = 105$ oocytes; 4E-T^Abl TRIM + Flag-4E-T^TRIM, $n = 104$ oocytes). Percentage of oocytes with GVBD spots is given as mean ± s.d. from four independent biological replicates. p-values were calculated using one-way ANOVA with Tukey's multiple comparisons test. **c** Mouse oocytes were injected with the indicated 4E-T or unspecific control (Ctrl) antibodies and mRNA encoding mouse TRIM21. Samples were kept in medium containing 10 μM RO-3306 for 6 h after injection and immunoblotted as indicated. One representative experiment of two independent replicates is shown. **d** Representative stills from time-lapse movies of control and 4E-T-depleted mouse oocytes stained with 5-SiR-Hoechst. Oocytes were incubated with dbcAMP as indicated. Cyan, DNA. Arrowheads indicate oil droplets injected to ensure quantitative microinjections. **e** Percentage of control and 4E-T-depleted mouse oocytes that resume meiosis (upper panel) or extrude polar bodies (lower panel) in the presence or absence of dbcAMP. Data are from four (upper panel) or two (lower panel) independent

experiments. **f** TRIM-Away of 4E-T and expression of Flag-4E-T^TRIM was performed as described in (**b**). 22 h after the second injection, oocytes were lysed for immunoblotting as indicated. One representative experiment of three independent biological replicates is shown. **g** Oocytes from (**f**) (Ctrl TRIM + H$_2$O, $n = 76$ oocytes; 4E-T^Abl TRIM + H$_2$O, $n = 68$ oocytes; 4E-T^Abl TRIM + Flag-4E-T^TRIM +, $n = 75$ oocytes; 4E-T^Abl TRIM + Flag-4E-T^TRIM ++, $n = 72$ oocytes) were treated with PG 22 h after the second injection. Percentage of oocytes undergoing GVBD in 760 min after PG addition and median time to GVBD are given as mean±s.d. from three independent biological replicates. $p$ values were calculated using one-way ANOVA with Tukey's multiple comparisons test. **h** Upper panel: Scheme of c-Mos expression, Cdk1 and MAPK activation as shown by loss of inhibitory and gain of activating phosphorylation, respectively, during prophase-I arrest and oocyte maturation. TRIM-Away of 4E-T and expression of Flag-4E-T^TRIM was performed as described in (**b**). 22 h after the second injection, oocytes were treated with PG, samples were taken at the indicated time points and immunoblotted as indicated. One representative experiment of three independent biological replicates is shown. **i, j** Experiment was performed with the indicated Flag-4E-T^TRIM constructs as described in (**f, g**). Oocytes (Ctrl TRIM + H$_2$O, $n = 105$ oocytes; 4E-T^Abl TRIM + H$_2$O, $n = 111$ oocytes; 4E-T^Abl TRIM + Flag-4E-T^TRIM, $n = 94$ oocytes; 4E-T^Abl TRIM + Flag-4E-T^TRIM 1-319, $n = 101$ oocytes) from four independent biological replicates were analyzed. Percentage of oocytes undergoing GVBD in 460 min after PG addition and median time to GVBD are given as mean±s.d. $p$ values were calculated using one-way ANOVA with Tukey's multiple comparisons test. Source data including additional loading controls are provided as a Source Data file.

we again depleted 4E-T from oocytes that we kept in the presence of the Cdk inhibitor Roscovitine to ensure that any effect on translation is not an indirect consequence of Cdk1 activation. As expected, we observed a very strong increase in cyclin-B1 and a more modest increase in c-Mos protein levels, both of which could be rescued by expression of ectopic 4E-T (Fig. 2e). The limited accumulation of c-Mos is probably due to the complete lack of stabilizing Cdk1 phosphorylations[37], thus resulting in a constantly high degradation under the selected assay conditions. Next, we aimed to test if the observed increases in cyclin-B1 and c-Mos protein upon 4E-T depletion contribute to spontaneous, PG-independent meiotic resumption. To suppress cyclin-B1 translation, we injected reported antisense oligonucleotides into the oocytes[38] (Fig. S2a). To prevent activation of the MAPK pathway as a consequence of c-Mos expression, we treated the oocytes with the MEK inhibitor U0126. Similar to previous reports on PG-induced meiotic resumption[39], we observed that suppression of either cyclin-B1 expression or MAPK activation significantly delayed but did not completely abrogate spontaneous induction of GVBD (Fig. S2b, c). From these experiments, we concluded that multiple deregulated 4E-T targets convey the effects of its depletion.

Based on our data, we speculated that 4E-T in response to PG dissociates from c-Mos, cyclin-B1 and cyclin-A1 mRNAs to allow their expression. To test this, we immunoprecipitated 4E-T from oocytes that were either (1) prophase-I-arrested ($t = 0$ h), (2) treated with PG but before GVBD ($t = 3$ h), or (3) treated with PG and after GVBD ($t = 6$ h). Intriguingly, 4E-T clearly dissociated from the c-Mos mRNA already before Cdk1 activation ($t = 3$ h), while cyclin-B1 and cyclin-A1 mRNAs were released only after GVBD ($t = 6$ h) (Fig. 2f). Taken together, we concluded that 4E-T is critical for translational repression of key M-phase promoting factors and its dissociation from distinct mRNAs is temporally controlled, which in the case of c-Mos and cyclin-B1 mRNAs is sufficient for translational derepression (Fig. 2g).

To gain insights into further molecular pathways controlled by 4E-T, we aimed to analyze how the depletion of 4E-T affects the oocytes' proteome. To this end, we used TRIM-Away to deplete 4E-T from immature oocytes for 42 h (Fig. S3a). Ctrl-depleted oocytes served as control. Throughout the whole experiment, all oocytes were treated with the Cdk inhibitor Roscovitine to maintain the prophase-I arrest and, thus, to prevent indirect changes in the translational state of mRNAs induced by Cdk1 activation during meiotic maturation[40]. The total proteome of Ctrl- and 4E-T-depleted

oocytes from three independent biological replicates was analyzed by LC-MS/MS. Overall, 5565 proteins were reliably identified and quantified. As expected, both the L- and S-version of 4E-T were found to be significantly decreased in the 4E-T-depleted oocytes (Fig. S3b). Of note, despite repeated attempts, we could not reliably detect unambiguous peptides of cyclin-B1 and c-Mos in any of the samples, even though our immunoblot analyses revealed strong upregulation of cyclin-B1 and to a lesser extent of c-Mos upon 4E-T depletion (Fig. S3a). Irrespective of that, other critical regulators of meiosis and cell division were found to be upregulated upon depletion of 4E-T, suggesting that 4E-T controls their translation in oocytes. Among these proteins, we identified Rcc1, Kif2C and TPX2, which are involved in the regulation of the bipolar microtubule spindle required for chromosome segregation in oocytes[41–44]. To confirm this result by an independent approach, we selected TPX2 and created a reporter mRNA containing its mRNA 3´UTR, as explained above. Indeed, translation of this reporter mRNA, but not of the control reporter fused to the HBG1 3´UTR, was significantly upregulated in oocytes depleted of 4E-T and this could be fully rescued by expression of TRIM-resistant ectopic 4E-T (Fig. S3c, d). In addition, our MS analyses revealed that the levels of Geminin and Cdc7 were increased in 4E-T-depleted compared to control oocytes (Fig. S3b). Interestingly, recent MS-based analyses showed that both proteins are upregulated during physiological, hormone-induced meiotic maturation[45]. Thus, these results indicate that Geminin and Cdc7 mRNAs are negatively controlled by 4E-T during the prophase-I arrest, and it is tempting to speculate that they are liberated from 4E-T-mediated inhibition following progesterone stimulation. Such a mechanism would allow the oocyte to prepare its proteome for faithful replication of the genome during the rapid early embryonic divisions following fertilization[46,47]. Similarly, TRIM-Away of 4E-T led to the significant upregulation of several histone proteins (Fig. S3b). Interestingly, it has been shown that the levels of maternally supplied histone proteins are critical regulators of cell cycle duration and onset of zygotic genome activation in early fly and frog embryos[48,49]. The finding that 4E-T might control histone expression is supported by the recent finding that its interactor eIF4E1b, a cap-binding protein specifically associated with repressed mRNAs, is strongly bound to many histone mRNAs in early zebrafish embryos[50]. In sum, these data suggest that 4E-T is involved in the regulated translation of mRNAs encoding proteins with diverse cellular functions demanding

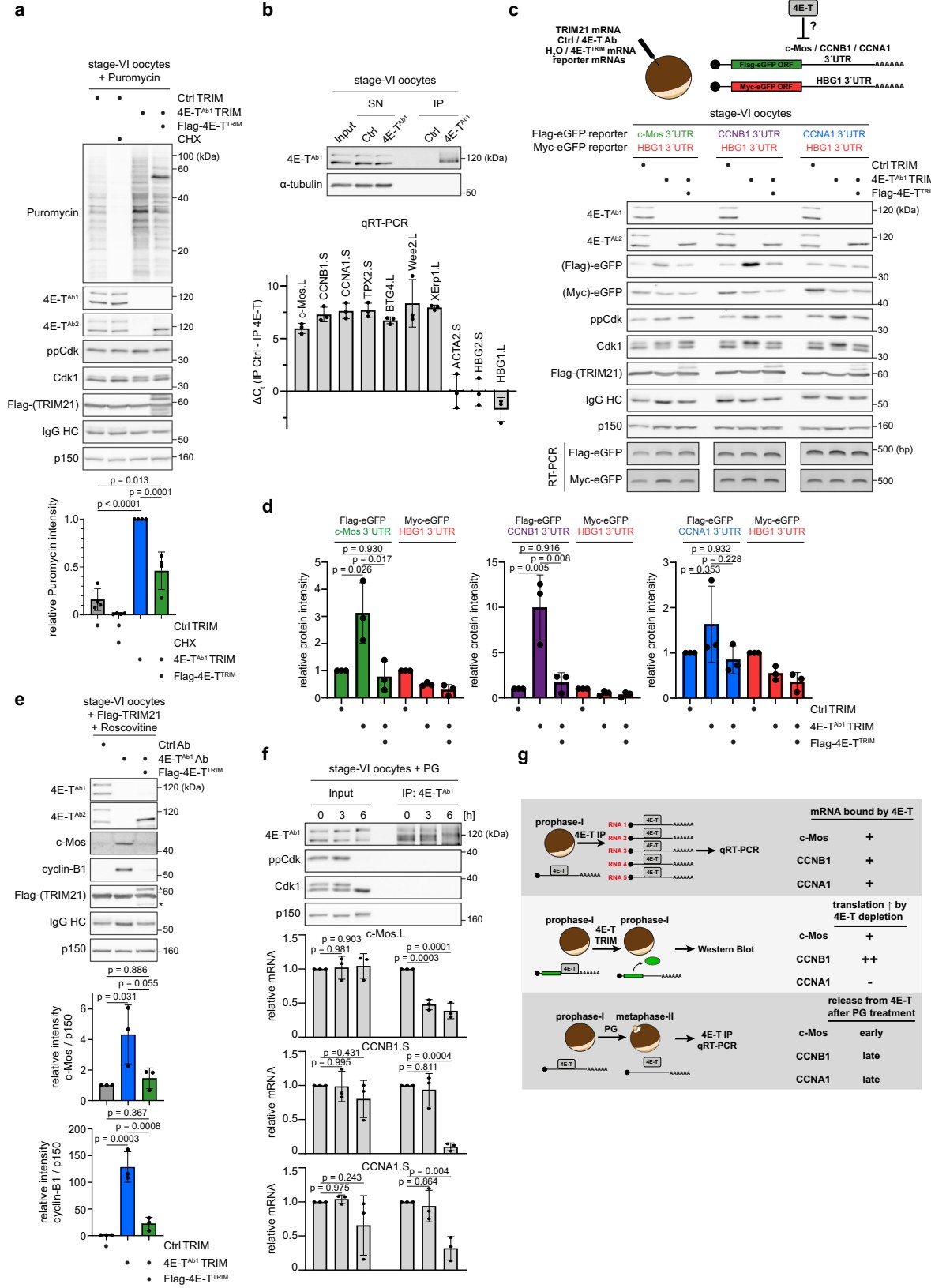

future studies to obtain a comprehensive picture of the complex function of 4E-T in oocytes.

## Characterization of meiotic RNPs containing 4E-T

To understand the molecular function of 4E-T, we next followed a three-step strategy: First, we analyzed the association of endogenous 4E-T with interacting proteins during meiotic maturation. Second, we analyzed 4E-T mutants to determine the regions mediating these interactions. Third, we harnessed this information to investigate which of those interactions are critical for 4E-T's function in intact oocytes.

In *Xenopus* oocytes, 4E-T is part of large RNPs containing well-known RNA-associated proteins like CPEB1, DDX6, PATL2, ePAB and

**Fig. 2 | 4E-T controls the translation of cell cycle mRNAs in Xenopus oocytes.**
**a** Xenopus stage-VI oocytes were injected with water or mRNA encoding Flag-4E-T[TRIM]. 18 h after injection, oocytes were co-injected with mRNA encoding Flag-TRIM21 and either 4E-T[Abl] or unspecific control (Ctrl) antibodies. 22 h after the second injection, oocytes were treated with puromycin and as indicated with cycloheximide (CHX) for 2 h. Oocytes were lysed and immunoblotted as indicated. Note, that we specifically selected oocytes that have not yet undergone spontaneous GVBD for analysis, to exclude indirect translational effects caused by pathways activated during meiotic maturation. Puromycin signals were quantified and normalized intensities from four independent biological replicates are given as mean ± s.d. p values were calculated using one-way ANOVA with Tukey's multiple comparisons test. **b** Xenopus stage-VI oocytes were lysed and subjected to immunoprecipitation with unspecific control (Ctrl) or 4E-T[Abl] antibodies. Samples were immunoblotted as indicated. In parallel, RNA was isolated from the same IP samples and analyzed by qRT-PCR for the indicated mRNAs. $\Delta C_t$ values between IP Ctrl and IP 4E-T[Abl] samples are given as mean ± s.d. from three independent biological replicates. **c** Xenopus stage-VI oocytes were injected with water or mRNA encoding Flag-4E-T[TRIM]. 18 h after injection, oocytes were co-injected with mRNA encoding Flag-TRIM21, with 4E-T[Abl] or unspecific control (Ctrl) antibodies, with mRNA encoding Myc-eGFP_HBG1 3′UTR and with mRNA encoding Flag-eGFP fused to the indicated 3′UTR. Oocytes were lysed after 22 h and immunoblotted as indicated. Note, that we specifically selected oocytes that have not yet undergone spontaneous GVBD for analysis, to exclude indirect translational effects caused by

pathways activated during meiotic maturation. One representative experiment of three independent biological replicates is shown. **d** Quantification of eGFP signals in (**c**). Values are given as mean ± s.d. from three independent biological replicates. p values were calculated using one-way ANOVA with Tukey's multiple comparisons test. **e** Xenopus stage-VI oocytes were injected with water or mRNA encoding Flag-4E-T[TRIM]. 18 h after injection, oocytes were co-injected with mRNA encoding Flag-TRIM21 and with 4E-T[Abl] or unspecific control (Ctrl) antibodies. Oocytes were incubated in medium containing the Cdk inhibitor Roscovitine. 48 h after the second injection, oocytes were lysed and immunoblotted as indicated. Cyclin-B1 and c-Mos signals were quantified and normalized to p150. Values were normalized to the Ctrl TRIM condition and are given as mean ± s.d. from three independent biological replicates. p values were calculated using one-way ANOVA with Tukey's multiple comparisons test. Asterisks indicate unspecific bands. **f** Xenopus stage-VI oocytes were treated with PG and lysed at the indicated time points. Lysates were subjected to immunoprecipitation with 4E-T[Abl]. Samples were immunoblotted as indicated. In parallel, RNA was isolated from the same samples and analyzed by qRT-PCR for the indicated mRNAs. Values in input and IP samples were normalized to t = 0 h conditions and are given as mean ± s.d. from three independent biological replicates. p values within Input and IP conditions were calculated using one-way ANOVA with Dunnett's multiple comparisons test. **g** Summary of results obtained for the c-Mos, CCNB1 and CCNA1 mRNAs. Source data including additional loading controls are provided as a Source Data file.

Zar family proteins[10–12,51,52]. After confirming these interactions by IP of endogenous 4E-T from stage-VI oocytes (Figs. 3a and S4a), we analyzed if these interactions are regulated. Therefore, we immunoprecipitated 4E-T from stage-VI oocytes (t = 0 h) and before (t = 3 h) or after (t = 6 h) GVBD following PG treatment. As indicated, the IPs were performed in the presence of RNaseA, and co-precipitating interactors were analyzed by WB (Fig. 3b). Next, we quantified these interactions to characterize the behavior of each individual protein upon the different treatment conditions (Fig. 3c). Interactions with eIF4E were partially RNA-dependent but did not change significantly during meiotic maturation. DDX6 and Myc-LSM14A co-precipitated with 4E-T in a highly RNA-dependent manner, but were unaffected by PG treatment. In contrast, interactions with PATL2, ePAB, Zar1l and Zar2 showed variable degree of RNA dependence, but were all highly sensitive to PG treatment and decreased during meiotic maturation. Specifically, interaction with PATL2 was only diminished after GVBD, i.e., after Cdk1 activation (at t = 6 h), whereas for the other three proteins reduced binding to 4E-T clearly preceded full Cdk1 activation (at t = 3 h). Of note, in the case of PATL2, Zar1l, and Zar2 decreased interactions with 4E-T were due to protein instability. Interactions with CPEB1 were mostly RNA-dependent with some variations between experiments (Figs. 3c and S4a). Importantly, its interaction decreased during meiotic maturation in a two-step manner, with a partial dissociation from 4E-T before Cdk1 activation (t = 3 h) and a full loss from 4E-T IPs after it (t = 6 h). The latter reduction was due to CPEB1 degradation. In contrast to reports on 4E-T in somatic cells[53], no or at least no strong interactions with CSDE1 (UNR) or CNOT1 were observed. The constant interaction between eIF4E and 4E-T suggests that it is not temporally regulated. However, since 4E-T dissociates from mRNAs during meiotic maturation (Fig. 2f), recruitment of 4E-T to distinct mRNAs might be mediated by specific adapter proteins, rather than exclusively via the eIF4E-5′ cap axis. Reportedly, PG-induced limited degradation of Zar1l before GVBD partially attenuates binding of CPEB1 to 4E-T[12]. Furthermore, PATL2 is degraded after GVBD suggesting that the initial remodeling triggered by Zar1l degradation is followed by a second, distinct mechanism affecting the composition of 4E-T RNPs later during meiotic maturation. The relevance of the 4E-T-PATL2 interaction will be studied in detail below.

Next, we analyzed the importance of 4E-T for RNP integrity. To this end, we separated Ctrl- or 4E-T-depleted stage-VI oocyte lysates by sucrose density gradient centrifugation (Fig. 3d). In Ctrl-depleted oocytes, 4E-T co-fractionated with its strong interactors, i.e., DDX6,

ePAB, Zar1l, Zar2, eIF4E, LSM14A and PATL2. Surprisingly, CPEB1 only partially co-fractionated, suggesting that a substantial pool of 4E-T RNPs contain mRNAs not regulated by CPEB1. Consistent with Fig. 3b, 4E-T did not co-fractionate with CSDE1 or CNOT1. Upon depletion of 4E-T, no altered fractionation pattern was observed for DDX6, ePAB, LSM14A and PATL2. In contrast, eIF4E was clearly less enriched in the high-density fractions. In summary, we propose that 4E-T – while not affecting overall RNP assembly – is required to stably assemble a translation repression complex that connects eIF4E at the 5′cap with large protein assemblies bound to the 3′UTR. Remodeling of this repressive complex during meiotic maturation (Fig. 3b, c) then coincides with the release of mRNAs encoding M-phase promoting factors from 4E-T (Fig. 2f), explaining why forced depletion of 4E-T triggers spontaneous GVBD (Fig. 1b).

## 4E-T directly interacts with eIF4E and PATL2

Next, we determined the 4E-T regions mediating the different interactions. Based on data on 4E-T in somatic cells[53], we deleted regions reported to interact with eIF4E (Δ4Ec (canonical) or Δ4Enc (non-canonical)), CSDE1 (ΔCSDE1), DDX6 (ΔDDX6, Cup homology domain), PATL (ΔPATL) and LSM14A (ΔLSM14A) and analyzed these mutants by IP from Xenopus stage-VI oocytes (Figs. 4a–c and S5a). As expected, the interaction with Myc-eIF4E was completely lost upon deletion of either eIF4E interaction region, but not affected by any other deletion. Deletion of the well-conserved Cup homology domain did not affect the interaction with DDX6 supporting our observation that binding of 4E-T to DDX6 is mainly RNA-dependent (Fig. 3b, c). The interaction with Myc-LSM14A, which was also almost completely RNA-dependent (Fig. 3b, c), was highly affected by ΔLSM14A and this also reduced interaction with other proteins that strongly rely on RNA to associate with 4E-T, like DDX6 and CPEB1. As expected, CSDE1 only very weakly co-precipitated with Flag-4E-T[TRIM] and this interaction—but not any of the other tested interactions—was abrogated by ΔCSDE1. In contrast, the ΔPATL deletion did not only diminish binding of Myc-PATL2, but also of all other tested factors except eIF4E, i.e. DDX6, LSM14A, Zar1l, CPEB1 and Zar2 (Figs. 4b, c and S5a), indicating that 4E-T-PATL2 form a critical hub for the assembly of a large ribonucleoprotein complex. From these data, we deduced that combining the Δ4Ec and ΔPATL deletions should completely abrogate binding of the tested interactors, which was indeed the case (Fig. 4d, e).

We then wondered if a 4E-T fragment (GFP-4E-T[447-563]) encompassing only the PATL interaction region was sufficient to mediate

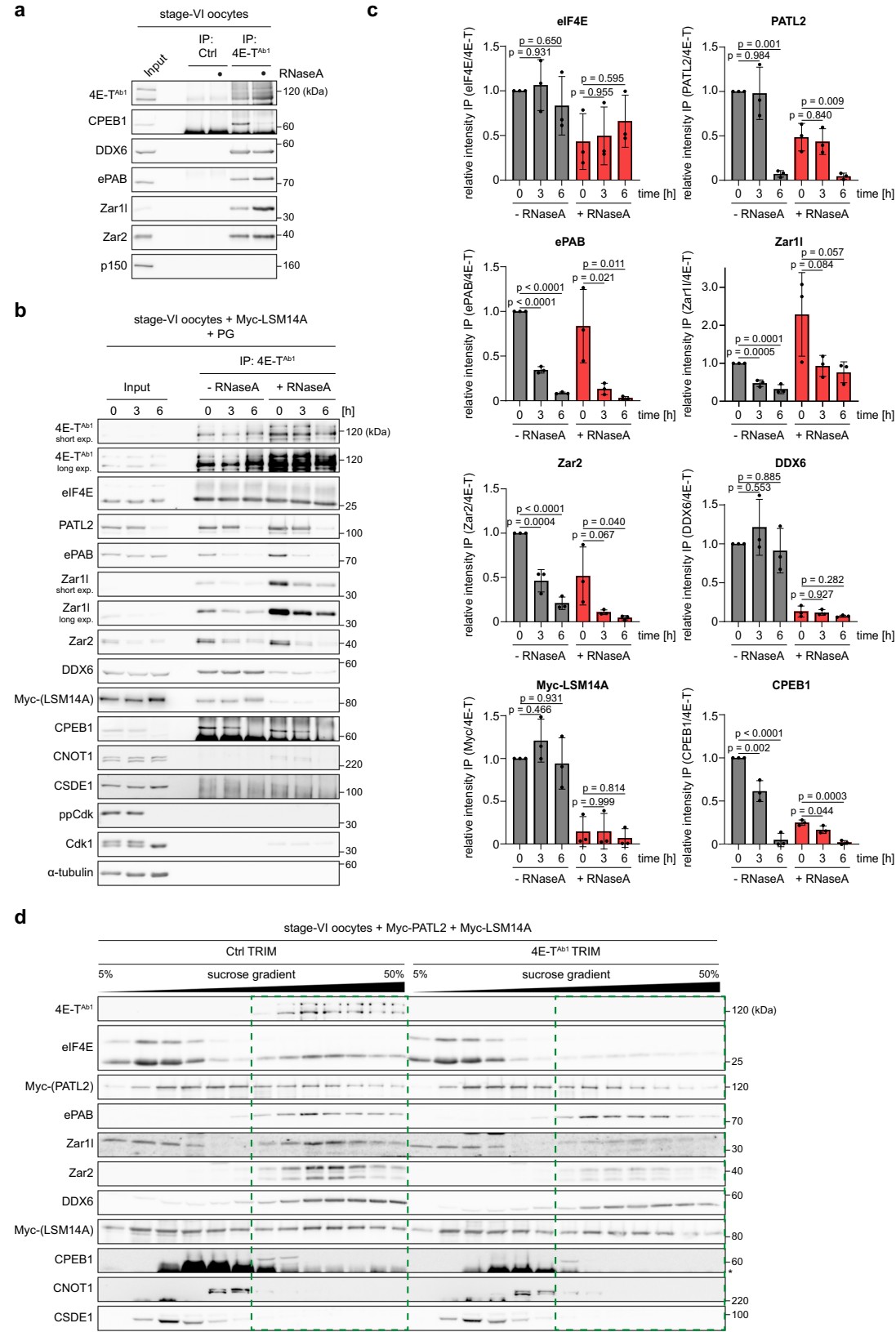

interaction with PATL2 and to recruit additional translation regulators. Ectopic PATL2 could be enriched by WT, but not ΔPATL, GFP-4E-T[447-563] from Xenopus stage-VI oocyte lysates (Fig. S4b). However, none of the other 4E-T interactors co-precipitated with the WT fragment. In summary, these assays demonstrate that in full-grown prophase-I-arrested Xenopus oocytes, 4E-T uses well-described regions to associate with the 5´cap-binding protein eIF4E and interacts via a conserved region with PATL2, which is necessary, but not sufficient, to form a complex with additional factors. It has been proposed that PATL proteins directly bind mRNAs, probably close to the 3´end[54,55], and it is tempting to speculate that PATL2-recruited 4E-T thus bridges the 5´cap with the 3´UTR to impose a stably inhibited closed mRNA

**Fig. 3 | Characterization of meiotic RNPs containing 4E-T. a** Xenopus stage-VI oocytes were lysed and treated with RNaseA as indicated. Lysates were subjected to immunoprecipitation with unspecific control (Ctrl) or 4E-T[Ab1] antibodies. Samples were immunoblotted as indicated. One representative experiment of three independent biological replicates is shown. Quantification of this figure can be found in Fig. S4a. **b** Xenopus stage-VI oocytes were injected with mRNA encoding Myc-LSM14A. 18 h after injection, oocytes were treated with PG and lysed at the indicated time points. Lysates were treated with RNaseA as indicated and subjected to immunoprecipitation with 4E-T[Ab1] antibodies. Samples were immunoblotted as indicated. One representative experiment of three independent biological replicates is shown. **c** Signals in IP samples from **(b)** were quantified and normalized to 4E-T signal in IP samples. Values were normalized to t = 0 h / −RNaseA condition and

are given as mean±s.d. from three independent biological replicates. *p* values within −RNaseA and +RNaseA conditions were calculated using one-way ANOVA with Dunnett's multiple comparisons test. **d** Xenopus stage-VI oocytes were injected with the indicated 4E-T or unspecific control (Ctrl) antibodies and mRNAs encoding Flag-TRIM21, Myc-PATL2 and Myc-LSM14A. 18 h after injection, oocytes were lysed and separated by sucrose density gradient centrifugation. Gradient fractions were analyzed by immunoblot as indicated. As reference, fractions containing 4E-T in Ctrl-depleted oocytes are shown as dashed rectangles in both conditions. Asterisk indicates IgG HC. One representative experiment of three independent biological replicates is shown. Source data including additional loading controls are provided as a Source Data file.

conformation. This would be reminiscent of other 4E-BPs, e.g. Drosophila Cup, which is recruited to mRNAs by the 3´UTR localized Bruno or Smaug proteins to repress mRNA translation[56].

### Interactions of 4E-T with eIF4E and PATL2 are required for its meiotic function

Following the third step of our strategy, we analyzed which of the 4E-T interactions are critical for its meiotic function. To this end, we depleted 4E-T from stage-VI oocytes, expressed either WT or mutant 4E-T[TRIM] and analyzed PG-induced meiotic maturation. We focused our analyses on the mutants with the strongest impact on 4E-T´s interaction network, i.e. the Δ4Ec, Δ4Enc, ΔPATL and ΔLSM14A variants of 4E-T. Compared to the non-rescue condition, the single-deletion mutants, except for the ΔLSM14A variant, were not able to significantly rescue timing of GVBD (Fig. 5a, b). Nevertheless, all tested constructs still showed a trend towards prolonged duration of meiotic resumption. We, thus, combined the Δ4Ec and ΔPATL deletions and analyzed the ability of the double deletion mutant to rescue meiotic timing in 4E-T-depleted oocytes. Notably, the Δ4Ec/ΔPATL double mutant did not rescue at all (Figs. 5c, d and S6a). Importantly, this mutant also failed to co-fractionate with endogenous 4E-T in dense RNPs of Xenopus stage-VI oocytes (Fig. 5e), suggesting that it is compromised in co-localizing with and binding to its target mRNAs. To test this, we immunoprecipitated either WT, single or double mutant 4E-T and quantified by qRT-PCR association with endogenous c-Mos, cyclin-B1 (CCNB1) and cyclin-A1 (CCNA1) mRNAs (Fig. 5f). 4E-T Δ4Ec associated weaker with all three mRNAs than WT, although in the case of CCNB1 and CCNA1, we observed substantial experimental variability. In contrast, the ΔPATL deletion consistently diminished association to all three mRNAs to a high extent. 4E-T Δ4Ec/ΔPATL was further compromised in mRNA binding. Of note, we also tested the ability of the ΔLSM14A mutant to bind to the c-Mos, CCNB1 and CCNA1 mRNAs, as this mutation strongly affects all tested RNA-dependent protein interactions of 4E-T (Figs. 4b, c and S5a), suggesting that the C-terminus of 4E-T could, directly or indirectly, be involved in RNA binding. Surprisingly, the ΔLSM14A mutant was only compromised in binding to the CCNB1 and CCNA1, but not the c-Mos, mRNAs (Fig. S6b). Thus, it is tempting to speculate that the C-terminus of 4E-T harbors sequence elements necessary to distinguish different target mRNA populations. In summary, these results strongly suggest that eIF4E and PATL2 are essential for independent mRNA recruitment of 4E-T, correct formation of repressive RNPs and the physiological function of 4E-T in prophase-I-arrested Xenopus oocytes.

### PATL2 is required to anchor 4E-T to its target mRNAs

Since PATL proteins localize to 3´UTRs of mRNAs[54] and 4E-T ΔPATL is compromised in target mRNA binding (Fig. 5f), we hypothesized that PATL2's main role is to anchor 4E-T to specific target mRNAs. If this applies, the interaction between 4E-T and PATL2 should become dispensable for translational repression if 4E-T is artificially tethered to an mRNA. To test this, we injected into Xenopus stage-VI oocytes a reporter mRNA containing the Myc-eGFP ORF and five boxB sites in its

3´UTR. The boxB sites are specifically bound by the λN-peptide[57], which we fused to 4E-T (Fig. 6a). Myc-dtTomato mRNA lacking boxB sites in its 3' UTR served as control. Importantly, co-expression of λN-Flag-4E-T WT strongly suppressed translation of the boxB reporter mRNA compared to water injected oocytes, while expression of control Myc-dtTomato mRNA was unaffected (Fig. 6b). In oocytes co-injected with λN-Flag-4E-T Δ4Ec, the tethered reporter mRNA was more strongly expressed than in 4E-T WT expressing cells, but significantly weaker than in water injected oocytes. Thus, in line with our previous data (Fig. 5b), preventing the interaction between eIF4E and eIF4G is an important−but not exclusive−mechanism by which 4E-T suppresses translation. Importantly, 4E-T ΔPATL suppressed translation of the boxB reporter mRNA to the same extent than 4E-T WT supporting the idea that PATL2's main function in Xenopus oocytes is to recruit 4E-T to target mRNAs rather than directly affecting translational repression. In summary, from these data we concluded that efficient translation repression by 4E-T critically depends on (1) the interaction with eIF4E to block eIF4G recruitment to the 5´cap and (2) the interaction with PATL2 to bind to specific target mRNAs.

## Discussion

We demonstrate that 4E-T is essential to maintain the long-lasting prophase-I arrest in full-grown Xenopus and mouse oocytes (Fig. 1b, e). Essential for this discovery was acute depletion of 4E-T in prophase-I-arrested oocytes using TRIM-Away without interfering with 4E-T's function during early oogenesis. Importantly, this allowed us to study the role of 4E-T specifically in full-grown prophase-I oocytes, without potential confounding effects that could occur if oocytes had undergone the preceding growth phase with reduced levels of this translational repressor. This critical distinction probably explains why earlier studies relying on RNAi or gene knockout strategies – lacking the temporal resolution of TRIM-Away – came to the seemingly contradicting observation that reduced levels of 4E-T result in oocyte maturation defects[18,24]. In line with its function as translation repressor in somatic cells[53], loss of 4E-T led to wide-spread upregulation of translation in oocytes (Fig. 2a). Among the activated mRNAs are key meiotic regulators, such as c-Mos and cyclin-B1 (Fig. 2d, e), suggesting that loss of 4E-T-mediated translational repression is sufficient to activate their expression to an extent that permits Cdk1 activation and, consequently, GVBD. Importantly, 4E-T dissociates from these mRNAs during meiotic maturation in unperturbed oocytes (Fig. 2f) strongly indicating that dynamic association of 4E-T controls their translation state and timing. PATL2 critically contributes to recruitment of 4E-T to target mRNAs (Fig. 5f), in line with reports that PATL proteins, either by themselves or by association with the Lsm1-7 ring, directly bind to RNA[55,58]. However, we have currently no evidence that PATL2 association is sufficient for the selective repression of distinct mRNAs by 4E-T. Interestingly, other proteins associated with 4E-T, e.g., CPEB1, DDX6, Zar1l and Zar2, have RNA binding capacity[12,59–61] and, therefore, dissecting their individual contribution to mRNA selectivity will be an interesting topic of future research. It is tempting to speculate that in oocytes there are different 4E-T RNPs with a defined set of core

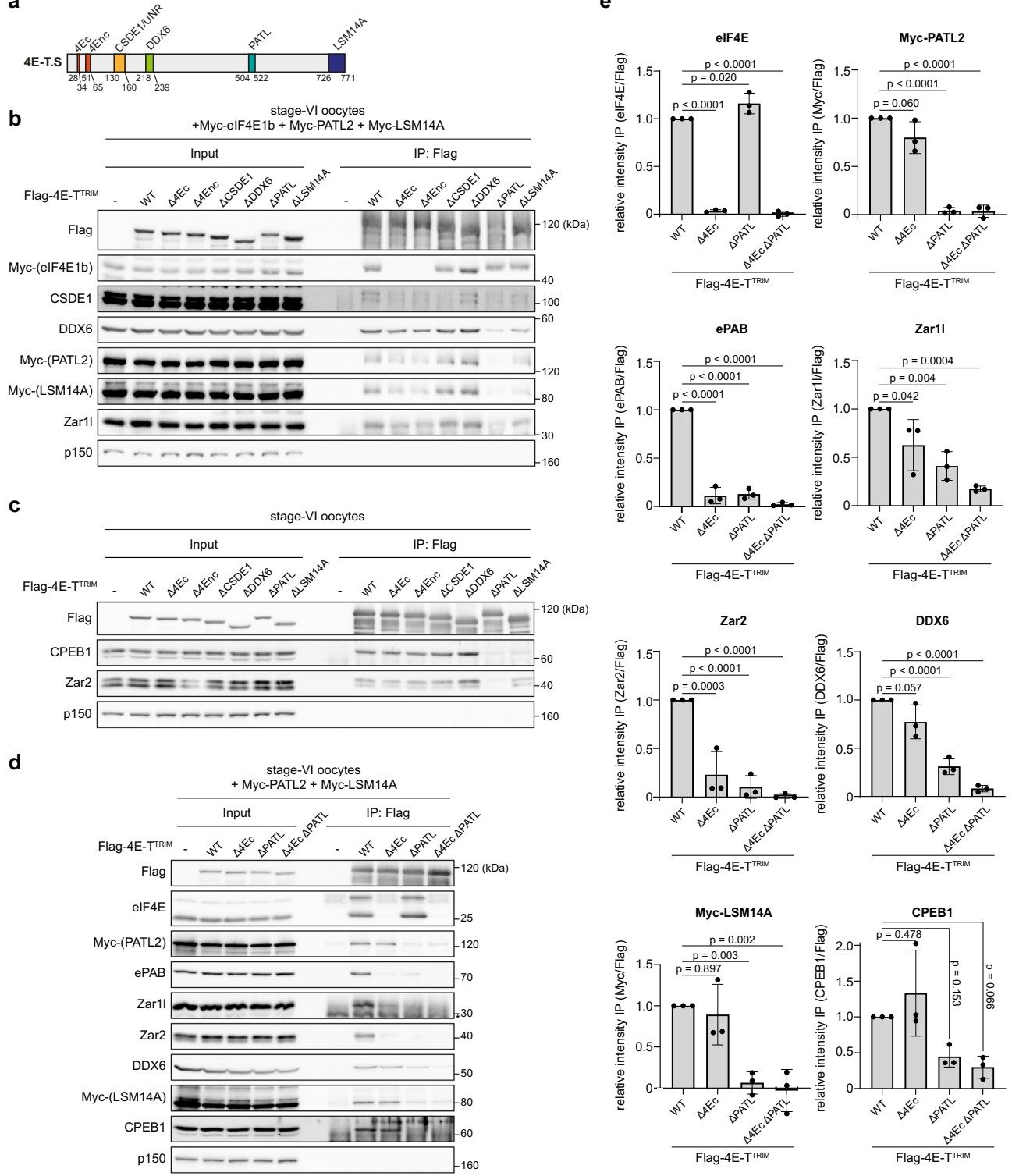

**Fig. 4 | 4E-T directly interacts with eIF4E and PATL2 in immature Xenopus oocytes. a** Schematic representation of 4E-T.S from *Xenopus laevis*. Regions required for interaction with the indicated proteins in somatic cells are highlighted[53]. **b** Xenopus stage-VI oocytes were injected with water or mRNA encoding the indicated Flag-4E-T$^{TRIM}$ variant and a mixture of mRNAs encoding Myc-eIF4E1b, Myc-PATL2 and Myc-LSM14A. 18 h after injection, oocytes were lysed and subjected to immunoprecipitation with Flag antibodies. Samples were immunoblotted as indicated. One representative experiment of three independent biological replicates is shown. Quantification of this figure can be found in Fig. S5a. **c** Xenopus stage-VI oocytes were injected with water or mRNA encoding the indicated Flag-4E-T$^{TRIM}$ variant. 18 h after injection, oocytes were lysed and subjected to immunoprecipitation with Flag antibodies. Samples were immunoblotted as indicated. One representative experiment of four independent biological replicates is

shown. Quantification of this figure can be found in Fig. S5a. **d** Xenopus stage-VI oocytes were injected with water or mRNA encoding the indicated Flag-4E-T$^{TRIM}$ variant and a mixture of mRNAs encoding Myc-PATL2 and Myc-LSM14A. 18 h after injection, oocytes were lysed and subjected to immunoprecipitation with Flag antibodies. Samples were immunoblotted as indicated. Asterisk indicates unspecific bands. One representative experiment of three independent biological replicates is shown. **e** Signals in IP samples from (**d**) were quantified. Signals in IP samples of water-injected oocytes were subtracted and values were normalized to Flag. All conditions were normalized to WT and values are given as mean±s.d. from three independent biological replicates. *p* values were calculated using one-way ANOVA with Dunnett's multiple comparisons test. Source data including additional loading controls are provided as a Source Data file.

proteins, e.g., eIF4E and PATL2, combined with mRNA-specific accessory proteins, e.g., Zar1l and Zar2, that bind specific mRNA sequence elements. Such variable RNP compositions could facilitate translational activation of distinct mRNAs with complex-specific dynamics, e.g., early and late after PG stimulation triggered by the degradation of Zar proteins and PATL2, respectively (Fig. 6c). These differences in the timing of degradation or posttranslational modifications of mRNA specific factors could explain the earlier dissociation of 4E-T from the c-Mos mRNA compared to the cyclin-B1 and cyclin-A1 mRNAs (Fig. 2f) and be leveraged by the oocyte to tailor translation profiles of individual mRNAs to developmental stage-specific needs.

In this study, we further map 4E-T's interaction network. Not surprisingly[53], interaction with eIF4E at the 5´cap via the known canonical and non-canonical binding sites is critical for 4E-T's meiotic function. Furthermore, binding of PATL2 is critical for the association with target mRNAs and additional factors like ePAB, Zar1l, Zar2, DDX6 or LSM14A (Figs. S5a and 5f). We thus conclude that the 4E-T-PATL2 interaction is at the heart of a large protein-protein interaction network that is important to bind to and, consequently, repress target mRNAs in oocytes. In the future, it will therefore be important to characterize the interface between 4E-T and PATL2 in detail and to determine if additional factors are required to stabilize this interaction. One potential candidate could be the Lsm1-7 ring, which has been shown to associate with PATL proteins[58]. The importance of the 4E-T-PATL interaction seems to be partially meiosis-specific, as in somatic cells deleting the same interaction region only abrogates binding to PATL1, but not DDX6 or LSM14A[53]. Binding of PATL2 is dispensable for translation inhibition if 4E-T is artificially tethered to an mRNA (Fig. 6b) suggesting that the main function of this interaction is indeed the stable recruitment of target mRNAs. We therefore predict that 4E-T's most critical interactions are with eIF4E, to compete for eIF4G binding at the 5´cap, and with PATL2, to efficiently bind target mRNAs in their 3´UTRs. Of note, during meiotic maturation the interaction between 4E-T and eIF4E is not weakened (Fig. 3b, c) implicating that it is rather mRNA binding via the PATL2-dependent complex that is regulated to adjust target mRNA translation.

An open question is the interplay between translation regulation by 4E-T and mRNA polyadenylation. Our data show that depletion of 4E-T is sufficient to induce translation of many proteins in full-grown prophase-I-arrested oocytes (Figs. 2a and S3a). As these oocytes receive no other stimulus, we postulate that at least some of these mRNAs are translationally upregulated in the absence of polyadenylation. In line with this, we found that CPEB1 co-fractionates only with a small part of the total cellular 4E-T (Fig. 3d), which suggests that only a subset of mRNAs controlled by 4E-T is also regulated by polyadenylation. In addition, it has been reported that prior to GVBD there is only a rather weak correlation between mRNA polyadenylation and increased polysome binding[4]. We thus propose that the role of other mechanisms, e.g., binding of inhibitory proteins such as 4E-T, has been underappreciated and that regulation of mRNA translation during prophase-I arrest and meiotic maturation is more complex than shortening and lengthening of polyA-tails.

## Methods

### Preparation of Xenopus laevis stage-VI oocytes
*Xenopus laevis* frogs were bred and maintained at the animal research facility, University of Konstanz, according to the regulations by the Regional Commission, Freiburg, Germany (Az. 35-9185.81/G-17/121 and 35-9185.81/G-22/080). Ovaries surgically removed from mature frogs were incubated in 1xMBS (5 mM HEPES pH = 7,8; 88 mM NaCl; 1 mM KCl; 1 mM MgSO₄; 2,5 mM NaHCO₃; 0,7 mM CaCl₂) supplemented with 50 ng/µl Liberase™ (Roche) at 23 °C for 90 min. After extensive washing in 1xMBS, stage-VI oocytes were collected and kept at 19 °C until further treatment.

### Preparation and culture of mouse oocytes
C57BL/6N mice were maintained at 21 °C ambient temperature, 52–55% humidity and a 14-h light/10-h dark cycle in a specific pathogen-free environment according to The Federation of European Laboratory Animal Science Associations guidelines and recommendations. Requirements of formal control of the German national authorities and funding organizations were satisfied, and the study received approval by the Niedersächsisches Landesamt für Verbraucherschutz und Lebensmittelsicherheit (LAVES). Full-grown oocytes were isolated from ovaries of 7- to 10-week-old mice and were kept arrested in prophase in homemade phenol red-free M2 medium supplemented with 250 µM dbcAMP under paraffin oil (NidaCon #NO-400K).

### *Xenopus laevis* oocyte lysis and time course experiments
Xenopus stage-VI oocytes were treated with 5 ng/µl PG (Sigma-Aldrich) in 1xOR2 (5 mM HEPES pH = 7,8; 82,5 mM NaCl; 2,5 mM KCl; 1 mM CaCl₂; 1 mM MgCl₂; 1 mM Na₂HPO₄) and incubated at 23 °C as indicated. To analyze meiotic timing, oocytes were imaged at 21 °C under a Stemi 2000-C (Zeiss) with a SPOT Insight™ 2MP Color camera. For immunoblotting, oocytes were lysed by mechanical shearing in 5 µl Lysis Buffer per oocyte (137 mM NaCl; 2,7 mM KCl; 10 mM Na₂HPO₄; 2 mM KH₂PO₄; 5 mM β-glycerophosphate; 2 mM NaF; 1x cOmplete™ Protease Inhibitor Cocktail (Sigma-Aldrich); pH = 7,4). For all immunoprecipitation, sucrose density gradient centrifugation and translation reporter experiments, the Lysis Buffer was additionally supplemented with 100 U/ml RNasin® Ribonuclease Inhibitor (Promega) or, when indicated, with 100 ng/µl RNaseA (Roth). Lysates were centrifuged at 20,000 × g for 10 min at 4 °C and the clear supernatant was retrieved. For immunoblot analysis, the clear supernatant was transferred to one volume of 3x Laemmli sample buffer (LSB).

### Microinjection of *Xenopus laevis* stage-VI oocytes
Needles for microinjection were pulled from glass capillaries (WPI #504949) with a P-97 Micropipette Puller (Sutter Instrument) and cut manually to the desired length. Typically, volumes between 9.2–18.4 nl were injected into stage-VI oocytes using a Nanoliter 2010 Micro-injection Pump (WPI).

### IVT and mRNA production
Coupled in vitro transcription and translation was performed with the TNT® SP6 High-Yield Wheat Germ Protein Expression System (Promega). mRNA was produced with the mMESSAGE mMACHINE™ T7 Ultra Transcription Kit (Thermo Fisher). After transcription, all mRNAs were polyadenylated with the polyA polymerase provided by the kit, except for the Flag-eGFP_c-Mos 3´UTR, Flag-eGFP_CCNB1 3´UTR, Flag-eGFP_CCNA1 3´UTR, Flag-eGFP_TPX2 3´UTR and Myc-eGFP_HBG1 3´UTR mRNAs.

### DNA constructs
DNA primers used for cloning can be found in the DNA oligonucleotides section. The following constructs were used in this study: Flag-4E-T (*Xenopus laevis* 4E-T.S, n-terminal 3xFlag-tag, wild-type); Flag-4E-T^TRIM (*Xenopus laevis* 4E-T.S, n-terminal 3xFlag-tag, Δ169-189); Flag-4E-T^TRIM 1-319 (*Xenopus laevis* 4E-T.S, n-terminal 3xFlag-tag, Δ169-189, amino acid Met1-Gln319); Flag-4E-T^TRIM Δ4Ec (*Xenopus laevis* 4E-T.S, n-terminal 3xFlag-tag, Δ169-189 Δ28-34); Flag-4E-T^TRIM Δ4Enc (*Xenopus laevis* 4E-T.S, n-terminal 3xFlag-tag, Δ169-189 Δ51-65); Flag-4E-T^TRIM ΔCSDE1 (*Xenopus laevis* 4E-T.S, n-terminal 3xFlag-tag, Δ169-189 Δ130-160); Flag-4E-T^TRIM ΔDDX6 (*Xenopus laevis* 4E-T.S, n-terminal 3xFlag-tag, Δ169-189 Δ218-239); Flag-4E-T^TRIM ΔPATL (*Xenopus laevis* 4E-T.S, n-terminal 3xFlag-tag, Δ169-189 Δ504-522); Flag-4E-T^TRIM ΔLSM14A (*Xenopus laevis* 4E-T.S, n-terminal 3xFlag-tag, Δ169-189 Δ726-771); Flag-4E-T^TRIM Δ4Ec ΔPATL (*Xenopus laevis* 4E-T.S, n-terminal 3xFlag-tag, Δ169-189 Δ28-34 Δ504-522); GFP-4E-T^447-563 (*Xenopus laevis* 4E-T.S, n-terminal eGFP-tag, amino acid Gln447-Gly563); GFP-4E-T^447-563 ΔPATL (*Xenopus laevis* 4E-T.S,

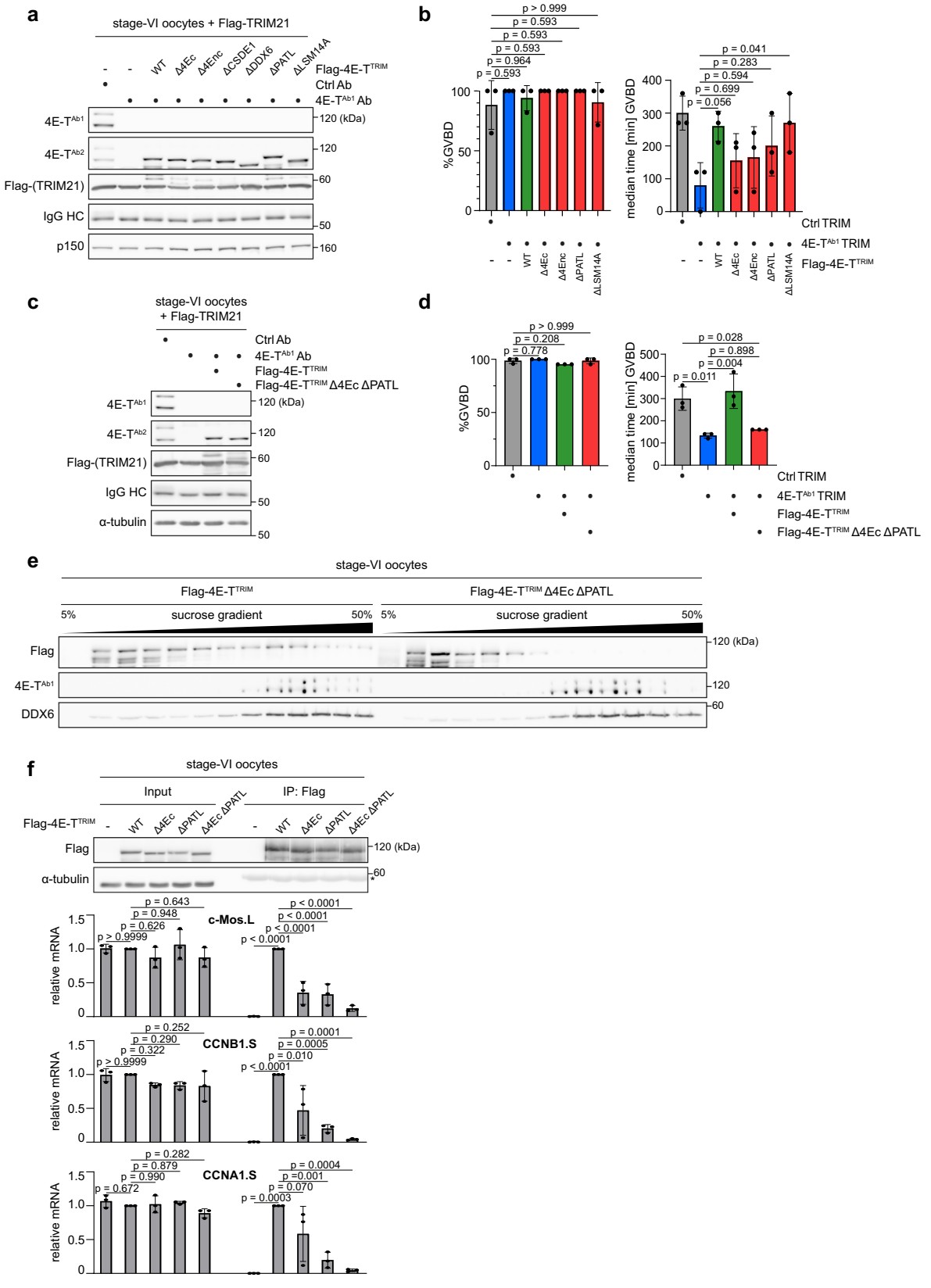

n-terminal eGFP-tag, Δ504-522, amino acid Gln447-Gly563); λN-Flag-4E-T^TRIM (*Xenopus laevis* 4E-T.S, n-terminal λN-Linker-3xFlag-tag, Δ169-189); λN-Flag-4E-T^TRIM Δ4Ec (*Xenopus laevis* 4E-T.S, n-terminal λN-Linker-3xFlag-tag, Δ169-189 Δ28-34); λN-Flag-4E-T^TRIM ΔPATL (*Xenopus laevis* 4E-T.S, n-terminal λN-Linker-3xFlag-tag, Δ169-189 Δ504-522); λN-Flag-4E-T^TRIM Δ4Ec ΔPATL (*Xenopus laevis* 4E-T.S, n-terminal λN-Linker-3xFlag-

tag, Δ169-189 Δ28-34 Δ504-522); Myc-PATL2 (*Xenopus laevis* PATL2.L, n-terminal 6xMyc-tag, wild-type); Myc-LSM14A (*Xenopus laevis* LSM14A.L, n-terminal 6xMyc-tag, wild-type); Myc-eIF4E1b (*Xenopus laevis* eIF4E1b.S, n-terminal 6xMyc-tag, wild-type); Flag-TRIM21 (*Mus musculus*, codon-optimized for *Xenopus laevis*, n-terminal 3xFlag-tag, amino acid Ser2-Met462, synthesized by Thermo Fisher); Myc-eGFP_5xboxB

**Fig. 5 | Interactions of 4E-T with eIF4E and PATL2 are required for its meiotic function. a** Xenopus stage-VI oocytes were injected with water or mRNA encoding the indicated Flag-4E-T$^{TRIM}$ variant. 18 h after injection, oocytes were co-injected with mRNA encoding Flag-TRIM21 and either 4E-T$^{Ab1}$ or unspecific control (Ctrl) antibodies. 22 h after the second injection, oocytes were lysed and immunoblotted as indicated. One representative experiment of three independent biological replicates is shown. **b** Oocytes from (**a**) (Ctrl TRIM + H$_2$O, n = 50 oocytes; 4E-T$^{Ab1}$ TRIM + H$_2$O, n = 51 oocytes; 4E-T$^{Ab1}$ TRIM + Flag-4E-T$^{TRIM}$, n = 51 oocytes; 4E-T$^{Ab1}$ TRIM + Flag-4E-T$^{TRIM}$ Δ4Ec, n = 53 oocytes; 4E-T$^{Ab1}$ TRIM + Flag-4E-T$^{TRIM}$ Δ4Enc, n = 53 oocytes; 4E-T$^{Ab1}$ TRIM + Flag-4E-T$^{TRIM}$ ΔPATL, n = 54 oocytes; 4E-T$^{Ab1}$ TRIM + Flag-4E-T$^{TRIM}$ ΔLSM14A, n = 59 oocytes) were treated with PG 22 h after the second injection. Percentage of oocytes undergoing GVBD in 390 min after PG addition and median time to GVBD are given as mean ± s.d. from three independent biological replicates. *p* values were calculated using one-way ANOVA with Dunnett's multiple comparisons test. **c** TRIM-Away of 4E-T and expression of the indicated Flag-4E-T$^{TRIM}$ variants was performed as described in (**a**). 22 h after the second injection, oocytes were lysed for immunoblotting as indicated. One representative experiment of three independent biological replicates is shown. **d** Oocytes from (**c**) (Ctrl TRIM + H$_2$O, n = 67 oocytes; 4E-T$^{Ab1}$ TRIM + H$_2$O, n = 64 oocytes; 4E-T$^{Ab1}$ TRIM + Flag-4E-T$^{TRIM}$, n = 63 oocytes; 4E-T$^{Ab1}$ TRIM + Flag-4E-T$^{TRIM}$ Δ4Ec ΔPATL, n = 62

oocytes) were treated with PG 22 h after the second injection. Percentage of oocytes undergoing GVBD in 600 min after PG addition and median time to GVBD are given as mean ± s.d. from three independent biological replicates. *p* values were calculated using one-way ANOVA with Tukey's multiple comparisons test. A detailed analysis of the GVBD timing in the different conditions can be found in Fig. S6a. **e** Xenopus stage-VI oocytes were injected with mRNA encoding Flag-4E-T$^{TRIM}$ or Flag-4E-T$^{TRIM}$ Δ4Ec ΔPATL. 18 h after injection, oocytes were lysed and separated by sucrose density gradient centrifugation. Gradient fractions were analyzed by immunoblot as indicated. One representative experiment of three independent biological replicates is shown. **f** Xenopus stage-VI oocytes were injected with water or mRNA encoding the indicated Flag-4E-T$^{TRIM}$ variant. 18 h after injection, oocytes were lysed and subjected to immunoprecipitation with Flag antibodies. Samples were immunoblotted as indicated. Asterisk indicates IgG HC. In parallel, RNA was isolated from the same samples and analyzed by qRT-PCR for the indicated mRNAs. Values in input and IP samples were normalized to the Flag-4E-T$^{TRIM}$ conditions and are given as mean ± s.d. from three independent biological replicates. *p* values within input and IP conditions were calculated using one-way ANOVA with Dunnett's multiple comparisons test. Source data including additional loading controls are provided as a Source Data file.

(eGFP, n-terminal 6xMyc-tag, 3´UTR containing 5xboxB sequences); Myc-dtTomato (dtTomato, n-terminal 6xMyc-tag); Flag-eGFP_c-Mos 3´UTR (3xFlag-eGFP followed by complete 3´UTR of *Xenopus laevis* c-Mos.L); Flag-eGFP_CCNB1 3´UTR (3xFlag-eGFP followed by complete 3´UTR of *Xenopus laevis* CCNB1.S); Flag-eGFP_CCNA1 3´UTR (3xFlag-eGFP followed by complete 3´UTR of *Xenopus laevis* CCNA1.S); Flag-eGFP_TPX2 3´UTR (3xFlag-eGFP followed by complete 3´UTR of *Xenopus laevis* TPX2.S); Myc-eGFP_HBG1 3´UTR (6xMyc-eGFP followed by complete 3´UTR of *Xenopus laevis* Hbg1.L).

## Antibodies

4E-T$^{Ab1}$ antibody was generated by immunizing rabbits with the peptide DRDVRGGEKDREPREGRDREKEYKDKRC and purification against the same peptide. 4E-T$^{Ab2}$ antibody was generated by immunizing rabbits with the peptide KSTGRRTPTVSSPVPGASFLQC and purification against the same peptide. PATL2 antibody was the hyper-immuneserum of a rabbit immunized with the peptides APKEEEPEALQPVKEAKGSEKAC and CPVIPPYTAVPS (Fig. S7a). c-Mos antibody was generated by immunizing rabbits with 2xStrep-SUMO-c-Mos aa247-359 and purification against the same protein (Fig. S7b). cyclin-B1 antibody was generated by immunizing rabbits with 2xStrep-SUMO-CCNB1.S aa1-102 and purification against the same protein (Fig. S7c). The following antibodies were purchased from commercial suppliers: Flag-tag antibody (Sigma-Aldrich F1804); CPEB1 antibody (Biozol MBS9213514); ppMAPK antibody (Cell Signaling #9106); MAPK antibody (Santa Cruz sc-154); Cdk1 antibody (Santa Cruz sc-54); p150 antibody (BD Transduction Laboratories 610473); DDX6 antibody (for Xenopus samples, Novus Biologicals NB200-191); DDX6 antibody (for mouse samples, Abcam ab174277); GFP antibody (Thermo Fisher MA5-15256); Puromycin antibody (Sigma-Aldrich MABE343); eIF4E antibody (Cell Signaling #9742); CSDE1 antibody (Bethyl Laboratories A303-158A); CNOT1 antibody (Cell Signaling #30289); 4E-T antibody (for mouse samples, Thermo Fisher Scientific PA5-51680); DDB1 antibody (Abcam ab109027); rabbit control antibody for TRIM-Away in Xenopus oocytes (Biozol GSC-A01008). rabbit control antibody for TRIM-Away in mouse oocytes (Merck Millipore 12-370). Myc (9E10) and α-tubulin (DSHB 12G10) antibodies were purified from hybridoma cells. ppCdk (phospho-Thr14 phospho-Tyr15) antibody was a gift from Tim Hunt. ePAB, Zar1l and Zar2 antibodies were described before[12]. HRP-coupled α-rabbit (#711-005-152) and α-mouse (#115-035-146) secondary antibodies were purchased from Dianova. In Figs. 2b, f and 3a, b, all primary antibodies derived from rabbit as well as cyclin-B1 blots in Figs. 1h and 2e were detected using a conformation-specific secondary antibody purchased from Cell Signaling (#5127).

## DNA oligonucleotides

DNA oligonucleotides were ordered from biomers.net. Sequences are provided in Supplementary Table 1.

## TRIM-Away of 4E-T in Xenopus oocytes

Xenopus stage-VI oocytes were injected with 4.1 ng mRNA encoding Flag-TRIM21 and 19.3 ng rabbit control or α−4E-T antibodies. Unless otherwise stated, oocytes were incubated in 1xMBS at 19 °C for 22 h. For rescue experiments, 9.2 ng of the indicated mRNAs were injected 18 h prior to injection of TRIM-Away reagents and oocytes were incubated at 19 °C in 1xMBS. As indicated, oocytes were treated with 200 μM Roscovitine (Calbiochem) or 50 μM U0126 (Promega). cyclin-B mRNA knock-down was performed using DNA antisense oligonucleotides described before[38]. To this end, 6.25 ng cycB5-2 sense or antisense oligonucleotides and 18.75 ng cyc8 sense or antisense oligonucleotides were co-injected into stage-VI oocytes with the TRIM-Away reagents.

## TRIM-Away of 4E-T in mouse oocytes

Ultrafiltration (Merck Millipore UFC505024) was performed to change the buffer of 4E-T antibody (Thermo Fisher Scientific PA5-51680) and control IgG (Merck Millipore 12-370) to PBS. 2 pl of 0.5 μM mouse *Trim21* mRNA and 3 pl of 0.2 mg/ml 4E-T antibody containing 0.1% NP-40 were co-injected as previously described[22]. Injected oocytes were kept in M2 medium supplemented with 250 μM dbcAMP or 10 μM RO-3306. 5-SiR-Hoechst was used to label DNA[62].

## Puromycin incorporation assay

Depletion of 4E-T by TRIM-Away and expression of rescue constructs in Xenopus stage-VI oocytes was performed as described above. 22 h after injection of TRIM-Away reagents, oocytes were transferred to 1xMBS supplemented with 100 μM puromycin (InvivoGen) and as indicated with 50 ng/μl cycloheximide (Calbiochem). Oocytes were incubated at 23 °C for 2 h before they were lysed as described above.

## Immunoprecipitation of endogenous 4E-T

For immunoprecipitation of endogenous 4E-T, unspecific rabbit control or 4E-T antibodies were first coupled to Dynabeads$^{TM}$ Protein G (Thermo Fisher Scientific) according to the manufacturer's instructions. Subsequently, they were covalently cross-linked by incubating the antibody-coupled Dynabeads two times for 30 min with 10 mg/ml DMP (Sigma-Aldrich) in 0.2 M Boric Acid pH = 9.0. The Dynabeads were washed three times in 0.2 M Ethanolamine pH = 8.0 and then incubated two times for 1 h in 0.2 M Ethanolamine pH = 8.0. The

**a**

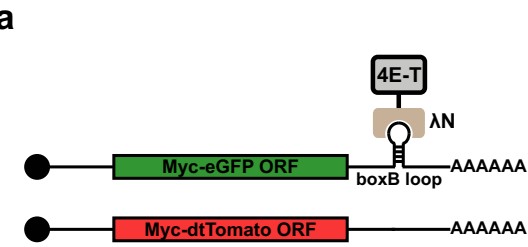

**b**

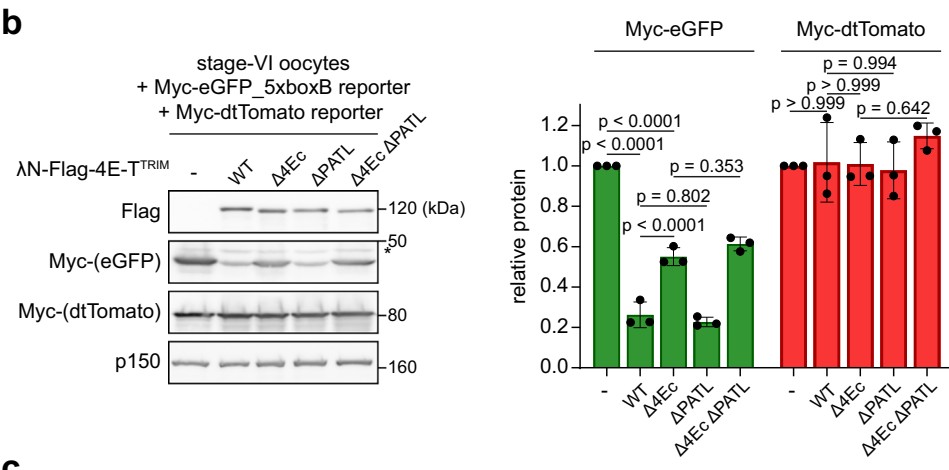

**c**

## prophase-I arrested oocyte

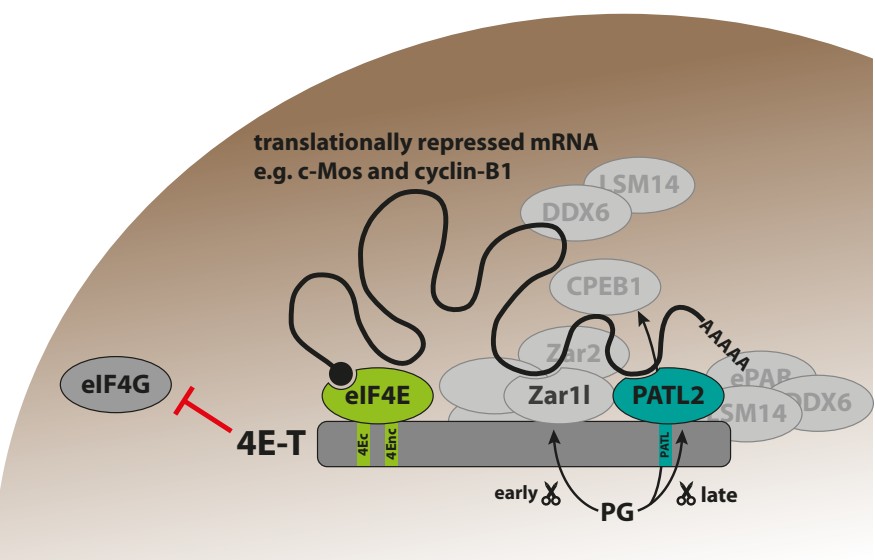

**Fig. 6 | PATL2 is required to anchor 4E-T to its target mRNAs. a** Schematic representation of the λN/boxB tethering assay in (**b**). **b** Xenopus stage-VI oocytes were injected with water or mRNA encoding the indicated λN-Flag-4E-T$^{TRIM}$ variants. 18 h after injection, oocytes were co-injected with a mixture of mRNAs encoding Myc-eGFP_5xboxB 3′UTR and Myc-dtTomato. 22 h after the second injection, oocytes were lysed and immunoblotted as indicated. Asterisk indicates unspecific bands. Myc signal of Myc-eGFP and Myc-dtTomato from three independent biological replicates was quantified. Values were normalized to water conditions and are given as mean ± s.d. *p* values were calculated using one-way ANOVA with Tukey's multiple comparisons test. **c** Working model for the function of 4E-T in suppressing mRNA translation in full-grown prophase-I-arrested oocytes by blocking access of eIF4G to eIF4E at the 5′ cap. The RNA-binding proteins Zar1l and PATL2 anchor 4E-T to selected target mRNAs. PG stimulation triggers early partial destruction of Zar1l (before GVBD) and late destruction of PATL2 (after GVBD). Source data including additional loading controls are provided as a Source Data file.

Dynabeads were washed three times with 0.1 M Glycine pH = 2.5 and three times with PBST. Xenopus stage-VI oocyte lysates were prepared as described above and 10 µl were added to 10 µl 3x LSB as input sample for immunoblotting. In Fig. 2b, e, 10 µl lysate were added to 150 µl QIAzol (QIAGEN) as input sample for qRT-PCR analysis. Of the residual lysates, 30 µl were added to the Dynabeads per µg cross-linked antibody. Per condition, 500 µl lysate were used in Fig. 2b, 200 µl lysate in Fig. 2f, 150 µl lysate in Fig. 3a and 230 µl lysate in Fig. 3b. After 1.5 h

incubation at 6 °C, the Dynabeads were washed three times with 1xWB Buffer (20 mM Tris; 300 mM NaCl; 0.2% Tween20; 1 mM EDTA; 1 mM EGTA; pH = 8.0). For experiments in Fig. 3a, b, the Dynabeads were resuspended in 1.5x LSB for immunoblot analysis. For experiments in Fig. 2b, f, 10% of Dynabeads were resuspended in 1.5x LSB for immunoblot analysis and 90% in 150 µl QIAzol (QIAGEN) for qRT-PCR analysis.

## Immunoprecipitation of ectopic 4E-T

Xenopus stage-VI oocytes were injected with water or 9.2 ng of the indicated Flag-4E-T mRNA. In Fig. 4b, oocytes were additionally co-injected with 1.7 ng Myc-PATL2 mRNA, 1.7 ng Myc-LSM14A mRNA and 1.7 ng Myc-eIF4E1b mRNA. In Fig. 4d, oocytes were additionally co-injected with 0.9 ng Myc-PATL2 mRNA and 0.9 ng Myc-LSM14A mRNA. Oocytes were incubated in 1xMBS at 19 °C for 18 h. Oocyte lysates were prepared as described above and 10 µl were added to 10 µl 3x LSB as input sample for immunoblotting. In Figs. 5f and S6b, 10 µl lysate were added to 150 µl QIAzol (QIAGEN) as input sample for qRT-PCR analysis. Of the residual lysate, 25 µl was added to Dynabeads™ Protein G (Thermo Fisher Scientific) per µg coupled α-Flag antibody. Per condition, 75 µl lysate were used in Fig. 4b, 50 µl lysate in Fig. 4c, 150 µl lysate in Fig. 4d and 200 µl lysate in Figs. 5f and S6b. After 1.5 h incubation at 6 °C, the Dynabeads were washed three times with 1xWB Buffer. For experiments in Fig. 4b–d, the Dynabeads were resuspended in 1.5x LSB for immunoblot analysis. For the experiment in Figs. 5f and S6b, 20% of Dynabeads were resuspended in 1.5x LSB for immunoblot analysis and 80% in 150 µl QIAzol (QIAGEN) for qRT-PCR analysis.

## qRT-PCR analysis

Total RNA was isolated from input and IP samples with RNeasy Mini Kit (QIAGEN). cDNA was synthesized with Transcriptor High Fidelity cDNA Synthesis Kit (Roche) using Random Hexamer primers. qRT-PCR was performed on a Light Cycler 96 (Roche) using the KAPA SYBR FAST for Light Cycler 480 Master Mix (KAPA Biosystems) with the following parameters: 3 min at 95 °C/45 cycles of 10 s at 95 °C, 20 s at 60 °C, 60 s at 72 °C. Specific primer pairs (see section "DNA oligonucleotides") were designed with Primer3Plus[63].

## Pull-down of GFP-tagged 4E-T

Xenopus stage-VI oocytes were injected with water or 8.3 ng of the indicated GFP-4E-T[447-563] mRNA. In addition, oocytes were co-injected with 0.9 ng Myc-PATL2 mRNA and 0.9 ng Myc-LSM14A mRNA. Oocytes were incubated in 1xMBS at 19 °C for 18 h. Oocyte lysates were prepared as described above and 10 µl were added to 10 µl 3x LSB as input sample for immunoblotting. Per condition, 140 µl lysate was added to 4 µl GFP-Trap Magnetic Agarose (Chromotek). After 1.5 h incubation at 6 °C, the beads were washed three times with 1xWB Buffer and resuspended in 1.5x LSB for immunoblot analysis.

## Sucrose density gradient centrifugation

5–50% sucrose density gradients with a total volume of 2 ml were prepared essentially as described before[64]. In Fig. 3d, Xenopus stage-VI oocytes were injected with 1.7 ng Myc-PATL2 mRNA, 1.7 ng Myc-LSM14A mRNA, 4.1 ng Flag-TRIM21 mRNA and 19.3 ng rabbit control or 4E-T[Ab1] antibody. In Fig. 5e, stage-VI oocytes were injected with 9.2 ng of the indicated Flag-4E-T mRNAs. Oocytes were incubated in 1xMBS at 19 °C for 18 h. Oocyte lysates were prepared as described above and 90 µl (Fig. 3d) or 190 µl (Fig. 5e) were carefully added onto the top layer of the sucrose gradient. Sucrose gradients were centrifuged with 51,000 rpm for 3 h in a TLS-55 rotor (Beckman Coulter) at 4 °C. From each gradient, 13 fractions were collected and added to one volume of 3x LSB.

## Translation reporter assay

The templates for the in vitro transcription of the translation reporter constructs Myc-eGFP_HBG1 3´UTR, Flag-eGFP_c-Mos 3´UTR, Flag-

eGFP_CCNB1 3´UTR, Flag-eGFP_CCNA1 3´UTR and Flag-eGFP_TPX2 3´UTR were generated by PCR using primers adding 30 adenylyl residues at the end of the 3´UTR. Depletion of 4E-T by TRIM-Away and expression of rescue constructs in Xenopus stage-VI oocytes was performed as described above. When injecting the TRIM-Away reagents, oocytes were co-injected with 0.5 ng Myc-eGFP_HBG1 3´UTR mRNA and as indicated with 0.5 ng Flag-eGFP_c-Mos 3´UTR mRNA, 0.25 ng Flag-eGFP_CCNB1 3´UTR mRNA, 0.5 ng Flag-eGFP_CCNA1 3´UTR or 0.5 ng Flag-eGFP_TPX2 3´UTR mRNA. Twenty-two hours after the second injection, oocytes were lysed as described above. Forty µl lysate was transferred to one volume of 3x LSB for immunoblot analysis. Ten µl lysate was transferred to 150 µl QIAzol (QIAGEN) for RT-PCR analysis. Total RNA was isolated with RNeasy Mini Kit (QIAGEN). cDNA was synthesized with Transcriptor High Fidelity cDNA Synthesis Kit (Roche) using Random Hexamer primers. PCR was performed using the PfuUltra II Fusion HS DNA Polymerase (Agilent) with the following parameters: 3 min at 96 °C/30 cycles (Myc-eGFP) or 35 cycles (Flag-eGFP) of 1 min at 96 °C, 30 s at 60 °C, 30 s at 72 °C/8 min at 72 °C. Primer pairs specific for the Flag-eGFP reporter (TGTTCTTTTTGCAGGATCCAC and GAACTTCAGGGTCAGCTTGC) and the Myc-eGFP reporter (TCTTTTTTGCAGGATCCCATC and GAACTTCAGGGTCAGCTTGC) mRNAs were designed with Primer3Plus[63].

## λN/BoxB tethering assays in Xenopus oocytes

The following plasmids were purchased from Addgene: BoxB in pcDNA3 was a gift from Howard Chang (Addgene plasmid #29727; http://n2t.net/addgene:29727; RRID:Addgene_29727)[65]. pCIneo-lambdaN_C was a gift from Elisa Izaurralde (Addgene plasmid #146024; http://n2t.net/addgene:146024; RRID:Addgene_146024). The λN-tag followed by a flexible linker was amplified by PCR from Addgene plasmid #146024 and inserted upstream of a 3xFlag-tag into a pCS2 plasmid suitable for in vitro transcription. Genes of interest can be introduced downstream of the λN-Linker-3xFlag-tag via Fse1 and Asc1 restriction sites. The 3´UTR containing five boxB sequences was amplified by PCR from Addgene plasmid #29727 and inserted downstream of a Fse1/Asc1 cloning site into a pCS2 plasmid encoding an N-terminal 6xMyc-tag that can be used for in vitro transcription. Xenopus stage-VI oocytes were injected with water or 7.8 ng mRNA encoding the indicated λN-Flag-4E-T variants and incubated in 1xMBS at 19 °C for 18 h. Oocytes were injected a second time with a mixture of 1.8 ng Myc-dtTomato mRNA and 1.8 ng Myc-eGFP_5xboxB 3´UTR mRNA. Oocytes were incubated in 1xMBS at 19 °C for 22 h before they were lysed as described above.

## Mass spectrometry analysis

Depletion of 4E-T by TRIM-Away in Xenopus stage-VI oocytes was performed as described above. Oocytes were incubated for 42 h in 1xMBS supplemented with 200 µM Roscovitine before they were lysed in 1xPBS as described above. 10 µl lysate were added to 10 µl 3xLSB for immunoblot analyses. A total of 100 µl lysate from three independent biological replicates were used for mass spectrometry analysis. Lysates were subjected to tryptic in-solution digest. Dried samples were resuspended in 8 M urea, followed by TCEP reduction (5 mM, 30 min, 37°, shaking at 650 rpm) and iodoacetamide alkylation (10 mM, 30 min, room temperature, shaking at 650 rpm) of cysteines. Samples were diluted to 1 M urea with 50 mM $NH_4HCO_3$ before digestion with 2.5 µg sequencing grade Trypsin/LysC (Promega) was performed over night for 18 h (37 °C, shaking at 650 rpm). Samples were detergent removed using prepacked HiPPR™ Detergent Removal Spin Columns (Thermo Scientific) according to manufacturer´s instructions with 100 µl settled resin per column and 50 mM $NH_4HCO_3$ buffer for equilibration. Proteolytic digestion was stopped by the addition of 2% formic acid and samples were desalted using Sep-Pak® Vac tC18 50 mg columns prior to MS measurement. Samples were measured at an

Orbitrap Eclipse™ Tribrid™ (Thermo Scientific) mass spectrometer equipped with a Vanquish™ Neo UHPLC (Thermo Scientific) system and a µPAC™ Neo HPLC column (length 50 cm, bed with 180 µm, interpillar distance 1.25 µm, pillar diameter 2.5 µm, pillar length 16 µm, pore size <100 Å, Thermo Scientific). The column was operated at 50 °C mounted in the column compartment. Peptides were separated at a flow of 300 nL/min, starting with 97% MS buffer A and 3% MS buffer B. This was followed by linear gradient increase to 30% MS buffer B for 219 min, increase to 45% MS buffer B for 10 min and 12 min washing at 100% MS buffer B. Data independent acquisition at the Orbitrap Eclipse™ Tribrid™ mass spectrometer was obtained in positive mode with a full MS scan at resolution 120,000, normalized AGC target 200%, max injection time 100 ms, scan range 350–1650 m/z, followed by 22 variable windows MS/MS scans at resolution 30,000, scan range 150–2000 m/z, charge state 2, RF lens 30%, AGC target 1000%, max injection time 54 ms and HCD normalized collision energy 28%. Samples were analyzed with Spectronaut® (Biognosys AG, v19.6.250122.62635). MS analysis was performed in directDIA™ mode with trypsin/P and LysC/P, and using MaxLFQ for label-free quantifications and the *Xenopus laevis* JGI v9.2 database downloaded from xenbase.org[66]. All other parameters were kept as standard setting. A false discovery rate of 1% was applied. Data were processed with Perseus (v2.1.3.0)[67]. Significantly enriched proteins were determined by an unpaired Student's *t* test (*n* = 3) and shown as volcano plot after log2 transformation using Python v3.10.0 and the pandas and numpy packages[68,69]. Volcano plot was created using R 4.4.2 with the ggplot2 and ggrepel packages[70,71].

## Data analysis

All GVBD timing and bar graphs were generated using GraphPad Prism 10. Band intensities in immunoblots were quantified using ImageJ. All values are given as mean±s.d. Statistical analyses were performed as indicated in the Figure Legends using GraphPad Prism 10. Mass spectrometry results were analyzed using Spectronaut®, Perseus, Python and RStudio. No statistical method was used to predetermine sample size. Oocytes were randomly assigned to experimental groups. Investigators were not blinded during data collection and analysis.

## Reporting summary

Further information on research design is available in the Nature Portfolio Reporting Summary linked to this article.

## Data availability

The mass spectrometry proteomics data have been deposited to the ProteomeXchange Consortium via the PRIDE[72] partner repository with the dataset identifier PXD062261. The authors declare that all other data generated or analyzed during this study are included in this published article (and its supplementary information files). Information and reagents required to repeat the experiments reported in this paper are available from the lead contact upon request. Source data are provided with this paper.

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

## Acknowledgements

The authors thank T. Hunt for the ppCdk antibody and Melanie Walter for assistance with frog surgeries. The work of A.H., J.O., F.S. and T.U.M. was funded by the Deutsche Forschungsgemeinschaft (CRC 969; B01) and the Konstanz Research School Chemical Biology of the Universität Konstanz (KoRS-CB). F.S. is grateful for funding by the DFG (project numbers 496470458 and 516836828). The work of S.C. and M.S. was supported by the Max Planck Society and the Deutsche Forschungsgemeinschaft (DFG, German Research Foundation) under Germany's Excellence Strategy (EXC 2067/1-390729940) and a DFG Leibniz Prize to M.S. (SCHU 3047/1-1).

## Author contributions

A.H. and T.U.M. conceived the study. A.H. performed all experiments in frog oocytes. S.C. performed 4E-T TRIM-Away experiments in mouse oocytes. J.O. and A.H. performed and analyzed mass spectrometry experiments. Figures were prepared by A.H., S.C. and T.U.M. The manuscript was written by A.H. and T.U.M. with input from all authors. M.S. supervised the studies in mouse oocytes and obtained funding. F.S. supervised mass spectrometry experiments and obtained funding. T.U.M supervised all other studies and obtained funding.

## Funding

## Competing interests

M.S. is a co-founder of Ovo Labs. All other authors declare no competing interests.
