## [Transparent Peer Review file · Nature Communications]

Translational repression by 4E-T is crucial to maintain the prophase-I arrest in vertebrate oocytes

Corresponding Author: Professor Thomas Mayer

Version 0:

Reviewer comments:

Reviewer #1

(Remarks to the Author)

The manuscript submitted by Heim et al. delves into the significance of 4E-T in maintaining the prophase-I arrest of vertebrate oocytes. Utilizing Trim-Away, the authors achieved an acute knockdown of 4E-T in mouse and frog oocytes. Their observations reveal alterations in the expression patterns of c-Mos and regulators of cyclin-1 in oocytes spontaneously released from prophase-I arrest. While these findings contribute to the field, several concerns must be addressed before considering the manuscript for publication in Nature Communications.

GENERAL ISSUES

1. Given the title "Translational repression by 4E-T is crucial to maintain the prophase-I arrest in vertebrate oocytes", the current scope of experiments limited to frog oocytes may not fully support the title's broader claim.
2. While 4E-T is well-known for its role in translation repression, the manuscript primarily focuses on its involvement in this process. However, a clearer understanding of how 4E-T specifically interacts with the translational repression complex and its unique function in vertebrate oocytes is lacking, limiting the study's novelty.

SPECIFIC ISSUES

1. The WB bands presented in the author's article are too faint, and the contrast needs to be adjusted. Additionally, the layout of the figures also requires optimization.
2. In Fig. 1d, the oocytes with 4E-T knockdown have extruded the first polar body, they also exhibit prominent pronuclear-like structures, suggesting that these oocytes have some defects in their MI.
3. In Fig. 1f, the WB results show the presence of two bands for endogenous 4E-T, but only one band for the rescued condition. What is the reason for this discrepancy?
4. In Fig. 1h, it can be seen that the rescue of ppMAPK is not as effective as that of c-Mos and ppCdk. What mechanism is involved in this difference? The authors have not provided any relevant description.
5. The mutation selected by the authors in Fig. 1i appears to be excessively severe, potentially affecting its functionality. It would be interesting to explore the consequences of replacing this mutation with other mutations, such as the one described in the article "A novel EIF4ENIF1 mutation associated with a diminished ovarian reserve and premature ovarian insufficiency identified by whole-exome sequencing".
6. The categorization of the four proteins in Fig. 3b and 3c requires further consideration. As the authors described in their article, the reduced interaction of PATL2, Zar11, and Zar2 is due to protein degradation, making it inappropriate to group them together with ePAB. Additionally, the interaction of CPEB1 may need further confirmation as the results in Fig. 3a and 3b are not well consistent.
7. The immunoprecipitation results in Fig. 4b should be quantitatively analyzed to provide a more intuitive interpretation of the Western blot (WB) data.
8. Based on the results in Fig. 4b-c, it is suggested to examine the binding ability of 4E-T-ΔLSM14A to mRNA, as this truncated form has reduced binding capacity for all mRNA-dependent proteins that interact with 4E-T.
9. Based on the results in Fig. 4b, it is evident that 4E-T-ΔDDX6 has minimal impact on the interaction of 4E-T with any other protein. Therefore, in Fig. 5b, it is not advisable to use this truncated form in the rescue experiment or to emphasize that it can fully rescue the phenotype, as this would be entirely predictable.
10. It is recommended to examine the expression of 4E-T binding partners in oocytes with 4E-T knockdown with WB.
11. Proteomic analysis of 4E-T knockdown oocytes would provide valuable insights into the molecular mechanisms involved.

12. RIP experiments for PATL2 are highly recommended to further elucidate its role and interactions within the cellular context.

13. Why were different control groups used in Fig. 5b and Fig. 5d? Was it because the experiments were performed in different batches? The trends seem to be somewhat inconsistent between the two groups of 4E-TAb1 TRIM + H₂O in Fig. 5b and Fig. 5d. Additionally, there appears to be some variability in the reproducibility between groups in Fig. 4b.

Reviewer #2

(Remarks to the Author)

In this manuscript, Heim et al. use the TRIM-away approach to demonstrate that acute depletion of the 4E-T produces spontaneous maturation in both mouse and *Xenopus* oocytes. They propose an interesting model to explain the well-established repression of the stockpiled mRNAs of Mos and Cyclin B1, two important regulators of meiotic division. This model involves the recruitment of 4E-T repression complex by Zar1L and PATL2 on the Mos and Cyclin B1 mRNAs in prophase I-arrested oocytes. During progesterone induced meiotic resumption, Mos mRNA is first released concomitantly with the degradation of Zar1L, while the Cyclin B1 mRNA is released later upon the degradation of PATL2.

Impact and novelty:

Understanding how translation is regulated during the prophase arrest and meiosis resumption is a critical question in the reproductive biology field. The acute depletion of 4E-T inducing meiotic resumption is an important and novel result providing new insight on how translation dormancy is maintained during the prophase arrest. The model about the control of translation during meiotic resumption proposed by the authors is also interesting. However, the involvement of the main actors, Zar1L and 4E-T, in the control of the prophase arrest and meiosis resumption is not completely original and was already published by the authors in a recent publication (10.1242/dev.200900). In this published paper, the authors have already described the interaction between 4E-T and Zar1L, and the degradation of the latter during Pg-induced meiosis resumption. Additionally, the interaction between 4E-T with PATL2 in repressing translation, and the degradation of the latter was already published by Nancy Standard in 10.1074/jbc.M704629200. More molecular insights on the mechanisms through which 4E-T induced meiosis resumption will strengthen the impact and the novelty of this study (See two key experiments in Figure 1).

Rational and design of the experiments:

The logic of the experiments is solid, the experimental plans is well designed and appropriate controls are used. However, the results presented in many panels look preliminary, since conclusions are often based on the qualitative observation of the single replicate shown, without quantification of the western blot signals and statistical analysis. Therefore, without a statistical analysis of the quantified signals, it is impossible to evaluate if all the conclusions of the authors are supported by the data (see figure by figure comments).

Major comments:

Figure 1.

In this figure the authors show that 4E-T acute depletion induces meiosis resumption in *Xenopus* and mouse oocytes. However, the molecular mechanisms through which this happens is only explored indirectly. Indeed, the authors show that the depletion of 4E-T accelerates meiosis resumption induced by progesterone by advancing the Mos accumulation, and potentially Cyclin B1. However, this experiment does not tell whether the acute reduction of 4E-T is sufficient to produce the translation de-repression of endogenous Mos or if the translation activation of endogenous Mos requires also progesterone. Additionally, it does not tell if the slow spontaneous maturation is due to the precocious accumulation of either Mos or Cyclin B1 or other unknown proteins. Since the 4E-T depletion is the most novel result of this paper, I think it deserves a more detailed experimental investigation, through two key experiments.

First, authors should investigate if acute 4E-T depletion in prophase oocytes induces the accumulation of endogenous Mos and Cyclin B1 proteins in the absence of progesterone. Since, 4E-T depletion induces meiotic maturation, Cdk1 activation has to be prevented with RO-3306 (similarly to what is done in the accompanying paper for DDX6 acute depletion), or by expressing a Cdk1 inhibitor as p21 (10.1091/mbc.10.10.3279) or Wee (PMID: 10673504).

Second, authors should test if blocking the translation of Mos or Cyclin B1 with antisense delays or blocks meiosis resumption induced by 4E-T. It was published that inhibiting the translation of both Mos and Cyclin B1 completely blocks progesterone-induced meiosis resumption (10.1038/sj.embor.7400611). It will be important to test if this is true when 4E-T is acutely depleted. This epistatic analysis is important to determine if 4E-T depletion induces meiosis resumption through the translation de-repression of Mos/Cyclin B1 without progesterone, or if it is through the translation de-repression of additional proteins, as for example Ringo/Speedy.

Fig. 1B: Statistical analysis is missing.

Fig. 1G: The representation of the meiotic maturation curves as mean and SD is confusing. This problem becomes more evident in figure 5B, where it is hard to visualize the data. The objective of these rescue experiments is to evaluate the time of meiosis resumption in the presence or absence of 4E-T. It would be more appropriate to compare the times of 50% GVBD among the different conditions. Furthermore, this will allow to perform a statistical analysis of the results essential for drawing any conclusions.

Fig. 1H: It would be important to evaluate the accumulation of endogenous Cyclin B1 in these oocytes. Since Cyclin B1 mRNA is also recruited to this inhibitory complex and Cyclin B1 overexpression is sufficient to induce meiotic resumption, it

might be an increase in Cyclin B1 accumulation the reason of the spontaneous maturation.

Fig. 1J: Similar problem than Fig. 1G.

Fig. 2A: Quantification of the puromycin signals on the different replicates of the experiment and statistical analysis are missing before concluding on the increase of global translation upon 4E-T acute depletion.

Fig. 2B: Interestingly, *wee2* mRNAs is included in the 4E-T inhibitory complex. It would be interesting to evaluate if *wee2* mRNA is translated upon 4E-T acute depletion. This would be important since *wee2* expression in prophase I could arrest/slow down meiosis resumption.

Fig. 2D: The statistical analysis is missing. The labeling of the Y-axis is confusing.

Fig. 2E: It will be important to show in the input lines, the abundance of Mos, Cyclin B1 and Cyclin A1 proteins by western blot. This will strengthen the observation reported showing that the timing of the protein accumulation matches the timing when their mRNAs are released from 4E-T. The statistical analysis of the mRNA quantifications is missing.

Fig. 3A-B: Some of the interaction looks very variable in the two illustrated replicates: CPEB1 in Fig.3A looks almost completely RNA-dependent, but in Fig 3B: CPEB1 interaction with 4E-T is instead almost completely resistant to RNAase treatment at 0h. In Fig 3B, it looks odd that Zar1L is recovered at lower level at 0h in the absence of RNAse treatment as compared to the RNAse treated sample. This last observation could be due to the lower efficiency of the 4E-T IP in samples not treated with RNAse. However, if the difference in the efficiency of 4E-T IP is taken in account, PATL2 would be higher at 0h in the sample not treated with RNAse as compared to the one treated, contradicting the conclusion of the authors that PATL2 interaction is RNAse independent.

The western blot signals need to be quantified in the different replicates of these experiments. The levels of the interactors of 4E-T should be normalized to the level of 4E-T IP-ed. A statistical analysis is critical to determine which interactions are dependent or not on RNA, and which interactions are stable or not during meiotic resumption. Without this analysis, it is difficult to reach any conclusions.

Fig. 3B: It was reported that there are multiple eIF4E expressed in oocytes (10.1074/jbc.M704629200,10.1101/gad.350400.123.,10.1038/s44319-023-00006-4). Interestingly, different eIF4E proteins bind to distinct set of mRNAs to produce translation activation or inhibition (10.1038/s44319-023-00006-4).

Is the antibody used in this study able to distinguish these different eIF4Es?

It would be important to evaluate which eIF4E isoform is included in this complex since they have dramatically opposite effects on translation.

Fig. 3C: This table is important to well summarize the data from the previous two experiments.

However, because of the qualitative approach used in Fig. 3A-B, at the current stage, this summary is not completely supported by the data.

Fig. 4B-C-D: The western blot signals need to be quantified in the different replicates of these experiments. The levels of the interactors of 4E-T should be normalized to the level of 4E-T IPed. A statistical analysis is critical to determine which interactions are taking place or not upon deletion of the different domains. Without this analysis, it is hard to drive conclusions.

Some of the contradictions between the replicates shown are:

In Fig. 4B, PATL2 and LSM14A binding is strongly decreased in 4Ec-mutant but almost unchanged in 4D.

Zar2 binding to 4Ec-mutant mutant seems unchanged in Fig. 4C but completely lost in Fig. 4D.

Fig. 5B is very confusing. Please see comments on Fig. 1G. Without a statistical analysis of the time of 50% GVBD, it is hard to assess if the conclusions of the authors are supported by the data. The plotting of the time of 50% GVBD for all conditions will also avoid repeating the control curves multiple times and reduce the complexity of the figure legend.

In Fig. 5F, the statistical analysis is missing to evaluate if the author conclusions are supported by the data.

In fig. 6B, the statistical analysis is missing to evaluate if the author conclusions are supported by the data.

Version 1:

Reviewer comments:

Reviewer #1

(Remarks to the Author)

Although the authors have revised the manuscript according the comments, the major concerns have not been solved.

1.The authors have verified the role of 4E-T in meiotic arrest in mice, however their observations seem to contradict previous findings: haploid loss of 4E-T leading to a decrease in GVBD rate (PMID: 38088064); knockdown of 4E-T by siRNA

resulting in the inability to undergo GVBD (PMID: 26147080). Furthermore, if the function of 4E-T is conserved in vertebrates, more experiments in mice are required for validation, especially regarding the repetition of important mechanisms (such as 4E-T's regulation of Mos and Cyclin-B1 mRNA).

2. Although the authors have performed many experiments on 4E-T, however the major findings have been reported previously. For example, CPEB1/4E-T/PATL2 complex and its inhibitory effect on translation and involvement in meiotic recovery have been previously reported by Minshall et al., 2007 (PMID: 17942399). Furthermore, the role of PATL2 as an RNA-binding component of mRNP (including 4E-T) in meiotic maturation has been documented: regulating the translation of c-Mos and Cyclin B1 mRNA, similar to the mechanisms in this study (PMID: 20471969, 20826699). Moreover, the regulation of c-Mos mRNA translation by PATL2 has also been reported in mice (PMID: 33614659).

3. Proteomic analysis of 4E-T knockdown oocytes and RIP experiments for PATL2 may give new insights into the molecular mechanisms and probably promote the novelty of the manuscript, however the authors have failure to address these comments.

4. The authors explained that incomplete restoration of translation in Fig. 2a is due to depletion of proteins interacting with 4E-T. Thus, protein expression in the rescue group should be added to Fig. S1c.

Reviewer #2

(Remarks to the Author)

I appreciate the work done by the authors to modify the figures and to perform the two additional experiments that strengthen the logic of this manuscript. The revised manuscript displays now quantifications and a statistical analysis of the experiments. I think these results would be interesting for the readership of Nature Communications.

However, two last concerns remain:

1) Authors used throughout the paper a series of pairwise t-test to compare multiple conditions of the same experiment, which is not appropriate. Indeed, the nature communication guidelines suggest: "Multiple comparisons: When making multiple statistical comparisons on a single data set, authors should explain how they adjusted the alpha level to avoid an inflated Type I error rate, or they should select statistical tests appropriate for multiple groups (such as ANOVA rather than a series of t-tests)."

An ANOVA test with multiple comparisons is advised to assess if the conclusions of the authors are statistically supported.

2) In figure 5b, the authors are displaying the comparison between the ability to rescue of the WT and the different mutants. Authors should instead compare the median time of the rescue conditions (WT or the mutants) with the median time of the TRIM-Away condition, which is the point of the rescue.

Version 2:

Reviewer comments:

Reviewer #1

(Remarks to the Author)

The authors have addressed my major concerns in the revisions.

Reviewer #2

(Remarks to the Author)

I thank the authors for having corrected the statistical analyses throughout the paper.

However, two important concerns remain, particularly regarding the mass spectrometry (MS) experiment the authors performed during the revision to evaluate differences between the proteomes of control and 4E-T-trimmed oocytes and Fig. 5.

a) Concerns regarding the MS experiment and data analysis:

1. Data analysis:

The use of Z-score normalization is suitable for pattern recognition (e.g., clustering of time resolved samples), but not for assessing differential expression between two conditions on three biological replicates. I recommend that the authors perform a statistical comparison by calculating log₂ fold changes and applying a significance test with FDR correction, presenting the results as a volcano plot or an equivalent. This will allow proper evaluation of the magnitude and statistical significance of observed differences.

2. Availability of raw data:

More raw MS counts should be provided as supplementary figures. Additionally, the raw data are still not available on PRIDE, as required.

3. Upon looking at the available data, several RNA-binding proteins (RBPs), including lsm14b.L, zar1.L, zar1.S, zar1.L, and patl2.L, appear less abundant in the TRIM condition. These changes could potentially contribute to the observed deregulation of translation and should be discussed if they are statistically significant. They could suggest a general destabilization of the complex that includes 4ET upon its removal.

4. Specificity of TRIM-away:

The clustering analysis shows that many ribosomal proteins are grouped into clusters 6 and 5 (the same clusters as 4E-T, which the authors have confirmed to be strongly depleted by TRIM-away). This is concerning and might indicate that the TRIM-away approach is depleting abundant proteins non-specifically, potentially affecting the general conclusions of the

study. Without statistical validation (as requested in point 1), it is not possible to assess whether the differences detected are real or experimental noise.

If statistical analysis confirms that some ribosomal proteins are significantly depleted, this must be discussed explicitly in the paper.

5. Validation of MS findings:

As also state by the authors, most proteins/mRNAs experimentally studied by the authors as deregulated by 4E-T TRIM are not detected in the MS dataset, as CcnB1 and Mos. The authors should validate at least some of the proteomic changes by western blot. Interestingly, TPX2 appears more abundant in TRIM samples; if this is statistically significant, it would support the 4E-T IP/qPCR findings presented.

Overall, without proper statistical analysis and independent validation (as by western blot), the following statements are not currently justified:

- “Mass spectrometry (MS) analyses identified additional proteins with diverse cellular functions to be upregulated in 4E-T depleted oocytes.”
- Lines 218–241.

b) Concerns regarding Figure 5b:

In Figure 5b, the correct statistical analysis shows no significant difference between the TRIM-away condition and its rescue with various 4E-T fragments (p-values ranging from 0.28 to 0.699), except for the LSM14 mutant ($p < 0.05$).

Therefore, the experiment must be repeated with more replicates, or the authors should revise their conclusion. For example, the statement:

- “Compared to the non-rescue condition, all tested single-deletion mutants were able to rescue meiotic timing to a certain extent with the Δ LSM14A variant being as efficient as WT 4E-T (Figs. 5a and 5b)” is not supported by the current data.

Version 3:

Reviewer comments:

Reviewer #2

(Remarks to the Author)

The authors have addressed my major concerns in the revision.

REVIEWER COMMENTS

We thank both reviewers for their positive and constructive comments. As shown by our point-by-point response, we addressed almost all the issues raised by the reviewers. We are grateful for the reviewers' insightful and positive comments because they have substantially improved our manuscript and the new data included in our revised manuscript significantly strengthen our findings. Please find below our point-by-point response addressing their specific comments.

Reviewer #1 (Remarks to the Author):

The manuscript submitted by Heim et al. delves into the significance of 4E-T in maintaining the prophase-I arrest of vertebrate oocytes. Utilizing Trim-Away, the authors achieved an acute knockdown of 4E-T in mouse and frog oocytes. Their observations reveal alterations in the expression patterns of c-Mos and regulators of cyclin-1 in oocytes spontaneously released from prophase-I arrest. While these findings contribute to the field, several concerns must be addressed before considering the manuscript for publication in Nature Communications.

We thank this reviewer for his/her positive comment and for appreciating the contribution of our findings for the field.

GENERAL ISSUES

1. Given the title “Translational repression by 4E-T is crucial to maintain the prophase-I arrest in vertebrate oocytes”, the current scope of experiments limited to frog oocytes may not fully support the title's broader claim.

Although we agree that most of the conclusions are drawn from experiments using *Xenopus* oocytes, the key finding that 4E-T is essential for the arrest of immature oocytes at prophase-I has been replicated in mouse oocytes (Fig. 1c-e). Thus, this function of 4E-T is conserved in at least two rather distantly related vertebrate species. We, therefore, think that the broader claim of the title is justified.

2. While 4E-T is well-known for its role in translation repression, the manuscript primarily focuses on its involvement in this process. However, a clearer understanding of how 4E-T specifically interacts with the translational repression complex and its unique function in vertebrate oocytes is lacking, limiting the study's novelty.

We respectfully disagree with the reviewer on this point and would like to justify our point of view using our findings on 4E-T and PATL2 as an example. Our study reveals that 4E-T interacts with the meiosis-specific PATL protein PATL2 (Fig. 2b). An interaction between 4E-T and PATL1 has been reported in somatic cells (Räsch et al., 2019, PMID: 32354837) and, therefore, one could argue that the interaction between 4E-T and PATL2 in oocytes is not surprising. However, PATL1 and PATL2 share less sequence conservation than it is to be expected given their similar names (31,6% sequence identity). We provide further molecular insights by demonstrating that the interaction between 4E-T and PATL2 is required for correct protein complex assembly (Fig. 4b-d and *new* Fig. 4e) and target RNA binding (Fig. 5f). In line with this, we provide evidence that the interaction with PATL2 is not required for translation repression of an mRNA in oocytes if 4E-T is artificially tethered to it (Fig. 6b). Importantly, we show that this interaction, together with the interaction with eIF4E at the 5' cap, is critical for the correct timing of progesterone-induced

meiotic resumption (Fig. 5d). Our study also reveals that molecular mechanisms can significantly differ between somatic and meiotic systems. For example, we unexpectedly discover that 4E-T's highly conserved Cup homology domain does not mediate interaction with DDX6 (Fig. 4b, Δ DDX6 4E-T), which is in stark contrast to the situation in somatic cells (Räsch et al., 2019, PMID: 32354837).

Thus, we believe that we not only revealed the essential function of 4E-T in maintaining the prophase-I arrest of full-grown oocytes, but also significantly advanced our understanding of its molecular mode-of-action in these oocytes. Key for these findings were our loss-of-function analyses in oocytes, which – to the best of our knowledge – have not been performed before. Importantly, these studies were possible because we acutely depleted 4E-T from full-grown oocytes using TRIM-Away without interfering with processes taking place during early oogenesis.

SPECIFIC ISSUES

1. The WB bands presented in the author's article are too faint, and the contrast needs to be adjusted. Additionally, the layout of the figures also requires optimization.

We agree with the reviewer that the layout of the figures required improvement. We incorporated new results and substantially changed the layout of the figures. We hope that these changes make the manuscript now easier to follow. Regarding the intensities of the WB bands, we tried our best to show our results as realistic as possible. This included that the intensities of the blots were selected so that each relevant band appears at most dark gray to ensure that differences in protein amounts are reflected in different gray scales. Yet, we agree that for some WB the bands were rather faint and, therefore, we replaced or complemented them with WBs showing longer exposure times (Figs. 1a, 1f, 1i, 3b, 3d, and 4d).

2. In Fig. 1d, the oocytes with 4E-T knockdown have extruded the first polar body, they also exhibit prominent pronuclear-like structures, suggesting that these oocytes have some defects in their MII.

We apologize for the confusion, but the pronuclear-like structure is actually an oil droplet. It's also visible in the control-depleted oocytes, although slightly out of focus. We usually perform quantitative microinjections in mouse oocytes to ensure experimental consistency. To this end, we always inject a small drop of oil into the oocyte as described previously (Kline, 2009, PMID: 19085140). Furthermore, we stained the DNA with Hoechst, and, therefore, if a pronuclear-like structure forms, it should be stained by the DNA dye. In the displayed oocytes, the chromosomes are labeled well, but the droplet is not stained. To avoid confusion, we have marked this structure as injection oil (**new** Fig. 1d).

3. In Fig. 1f, the WB results show the presence of two bands for endogenous 4E-T, but only one band for the rescued condition. What is the reason for this discrepancy?

We thank the reviewer for this comment. In *Xenopus*, a short and a long isoform of 4E-T are described (Minshall et al., 2007, PMID: 17942399), giving rise to two distinct bands in WB. These isoforms mostly differ by a 183 amino acid long stretch missing in the shorter isoform. In our hands, repeated attempts to amplify the longer isoform from cDNA failed and, therefore, we conducted the rescue experiments with the short version of 4E-T (NP_001086710.1). As shown in our study, expression of the short version is sufficient to rescue the loss of both isoforms (Fig. 1b).

4. In Fig. 1h, it can be seen that the rescue of ppMAPK is not as effective as that of c-Mos and ppCdk. What mechanism is involved in this difference? The authors have not provided any relevant description.

We thank the reviewer for pointing this out. In the two other biological replicates we performed, the increase in ppMAPK strongly correlates with that of c-Mos (see below). We thus believe that the inefficient rescue of ppMAPK observed in the original WB is due to the following reason: Although we generally observed a robust rescue of GVBD timing (Fig. 1b), upregulation of translation in oocytes depleted of 4E-T could not always be completely rescued (see Fig. 2A or **new** Fig. 2e). We believe that under these conditions, timing of c-Mos expression is slightly altered allowing premature accumulation of c-Mos. Yet, these low amounts of c-Mos were not detected in WB due to the low sensitivity of the c-Mos antibody (our experience). Low amounts of c-Mos result in MAPK activation detectable with the ppMAPK antibody, which in our hands is highly sensitive. In the revised version of our manuscript, we point out this aspect (line 136-140).

5. The mutation selected by the authors in Fig. 1i appears to be excessively severe, potentially affecting its functionality. It would be interesting to explore the consequences of replacing this mutation with other mutations, such as the one described in the article "A novel EIF4ENIF1 mutation associated with a diminished ovarian reserve and premature ovarian insufficiency identified by whole-exome sequencing".

We agree with the reviewer that the analyzed nonsense mutation alters the 4E-T protein more severely than missense mutations. Nevertheless, given the clinical relevance of the 4E-T nonsense mutation (Kasipillai et al., 2013, PMID: 23902945) we wanted to investigate to what extent it impairs the functionality of 4E-T. We also tested the mutation described in the publication mentioned by the reviewer, i.e., Q646P (corresponding to Q842P in human 4E-T (Zhao et al., 2019, PMID: 31810472)). But this mutation, like the one reported by Franca et al., 2020 (PMID:

33095795), efficiently rescued PG-induced meiotic timing in 4E-T-depleted oocytes (see below). Thus, we assume that mild mutations in 4E-T, that in the setting of a human oocyte have decades to evoke a phenotype, do not cause a measurable phenotype in our experiments, where oocytes are exposed to the mutation for only 2-3 days. An alternative explanation could be that these missense mutations exert their effect before oocytes reach the prophase I arrest. In this case, we would have missed a phenotype because we expressed mutant 4E-T in full-grown prophase I arrested oocytes acutely depleted of 4E-T.

6. The categorization of the four proteins in Fig. 3b and 3c requires further consideration. As the authors described in their article, the reduced interaction of PATL2, Zar11, and Zar2 is due to protein degradation, making it inappropriate to group them together with ePAB. Additionally, the interaction of CPEB1 may need further confirmation as the results in Fig. 3a and 3b are not well consistent.

We thank the reviewer for this comment. Taking up a similar comment raised by reviewer #2, we now quantified the abundance of all strong 4E-T interactors in the different IP conditions and replaced the initial table by the quantification results (**new** Fig. 3c). We agree that this provides a more holistic view on the individual interactions and complies more adequately with their complex nature (line 230-246).

We agree that there is a difference in the extent of RNaseA sensitivity in the CPEB1 interaction between Fig. 3a and Fig. 3b. We therefore repeated the experiment shown in Fig. 3b to obtain additional data from an independent biological replicate. Quantifications of these experiments (**new** Figs. 3c and S3a) confirmed that the CPEB1 interaction is more RNaseA sensitive under the experimental conditions of Fig. 3a compared to the ones of Fig. 3b. While we can only speculate about the reason for the difference in sensitivity, quantifications clearly show that under both experimental conditions the interaction with CPEB1 is significantly lost in large part upon RNaseA treatment. We rephrase the text accordingly to adequately meet this aspect (line 242–243).

7. The immunoprecipitation results in Fig. 4b should be quantitatively analyzed to provide a more intuitive interpretation of the Western blot (WB) data.

We agree with the reviewer and accordingly quantified the results of Fig. 4b-d. These quantifications are shown in the **new** Fig. 4e and **new** Fig. S4a. We also rephrase the text accordingly (line 289-300).

8. Based on the results in Fig. 4b-c, it is suggested to examine the binding ability of 4E-T Δ LSM14A to mRNA, as this truncated form has reduced binding capacity for all mRNA-dependent proteins that interact with 4E-T.

We thank the reviewer for this suggestion. We analyzed the amount of c-Mos, CCNB1 and CCNA1 mRNAs bound to ectopic WT or Δ LSM14A 4E-T immunoprecipitated from oocytes. Indeed, we observed that the Δ LSM14A mutation affected binding to CCNB1 and CCNA1 mRNAs (**new** Fig. S5b and line 338-345). Interestingly, we observed no significant reduction of binding to c-Mos mRNA. This could imply that the C-terminal stretch of 4E-T involved in LSM14A recruitment could be required to bind distinct mRNAs, an aspect that warrants further investigation in the future.

9. Based on the results in Fig. 4b, it is evident that 4E-T Δ DDX6 has minimal impact on the interaction of 4E-T with any other protein. Therefore, in Fig. 5b, it is not advisable to use this truncated form in the rescue experiment or to emphasize that it can fully rescue the phenotype, as this would be entirely predictable.

We agree with the reviewer and have removed the Δ DDX6 mutant from the original Fig. 5b. Following this rationale, we also removed the Δ CSDE1 mutant from the analysis because an interaction between 4E-T and CSDE1 was almost undetectable in oocytes and the Δ CSDE1 mutation did not significantly affect binding to any other tested protein (Figs. 3b, 3d, 4b and S4a). We agree that these changes enhance the focus on the relevant mutants we describe in this study.

10. It is recommended to examine the expression of 4E-T binding partners in oocytes with 4E-T knockdown with WB.

We agree with the reviewer and have analyzed the stability of the 4E-T binding partners upon 4E-T knockdown in oocytes (**new** Fig. S1c). We observed a significant reduction in the levels of Zar1l (to 0,40 \pm 0,11 of Ctrl KD), Zar2 (to 0,53 \pm 0,07 of Ctrl KD) and PATL2 (to 0,60 \pm 0,12 of Ctrl KD). In the revised version, we mention that this could be a reason why re-expression of just 4E-T does not always fully rescue the loss of endogenous 4E-T, e.g., in Fig. 2a (line 160-165).

11. Proteomic analysis of 4E-T knockdown oocytes would provide valuable insights into the molecular mechanisms involved.

We agree with the reviewer that a proteomic analysis of 4E-T knockdown oocytes would further advance our understanding of the function of 4E-T in oocytes. However, we feel that such an endeavor would exceed the scope of the current manuscript. The manuscript focuses on the essential function of 4E-T for the prophase-I arrest of immature oocytes. We identify cyclin-B1 and c-Mos as 4E-T targets, both of which are key drivers of oocyte maturation with each of them being sufficient to induce spontaneous, PG-independent meiotic resumption (Gaffre et al., 2011, PMID: 21795279; Yew et al., 1992, PMID: 1531698; Karaïskou et al., 2004, PMID: 14985258). Thus, we believe that a proteomic analysis, while being informative by itself, is not required to

understand the phenotype we investigated, but would rather provide the basis for future research directions.

12. RIP experiments for PATL2 are highly recommended to further elucidate its role and interactions within the cellular context.

Similar to the suggested proteomics approach (see point 11), we believe that a PATL2 RIPseq analysis, while being informative by itself, is not necessary to understand the investigated phenotype. In fact, such an analysis would likely pave the way for exploring novel avenues in future research endeavors.

13. Why were different control groups used in Fig. 5b and Fig. 5d? Was it because the experiments were performed in different batches? The trends seem to be somewhat inconsistent between the two groups of 4E-TAb1 TRIM + H2O in Fig. 5b and Fig. 5d. Additionally, there appears to be some variability in the reproducibility between groups in Fig. 4b.

The reviewer is right in that the oocytes used in Fig. 5b and Fig. 5d stem from different female frogs. It is a well-known phenomenon that oocytes from different frogs can differ significantly in their timing of meiotic maturation, as documented in studies such as Santoni et al., 2024 (PMID: 38358892). Thus, to reliably analyze the different conditions it was mandatory to include a control group for each oocyte batch. This variability in meiotic timing affects the comparability of results obtained from different female frogs as pointed out by the reviewer (see error bars in Fig. 5b, not 4b).

Reviewer #2 (Remarks to the Author):

In this manuscript, Heim et al. use the TRIM-away approach to demonstrate that acute depletion of the 4E-T produces spontaneous maturation in both mouse and *Xenopus* oocytes. They propose an interesting model to explain the well-established repression of the stockpiled mRNAs of Mos and Cyclin B1, two important regulators of meiotic division. This model involves the recruitment of 4E-T repression complex by Zar1L and PATL2 on the Mos and Cyclin B1 mRNAs in prophase I-arrested oocytes. During progesterone induced meiotic resumption, Mos mRNA is first released concomitantly with the degradation of Zar1L, while the Cyclin B1 mRNA is released later upon the degradation of PATL2.

We thank this reviewer for his/her positive comments and the appreciation of our findings.

Impact and novelty:

Understanding how translation is regulated during the prophase arrest and meiosis resumption is a critical question in the reproductive biology field. The acute depletion of 4E-T inducing meiotic resumption is an important and novel result providing new insight on how translation dormancy is maintained during the prophase arrest. The model about the control of translation during meiotic resumption proposed by the authors is also interesting. However, the involvement of the main actors, Zar1L and 4E-T, in the control of the prophase arrest and meiosis resumption is not completely original and was already published by the authors in a recent publication (10.1242/dev.200900). In this published paper, the authors have already described the interaction between 4E-T and Zar1L, and the degradation of the latter during Pg-induced meiosis resumption. Additionally, the interaction between 4E-T with PATL2 in repressing translation, and the

degradation of the latter was already published by Nancy Standard in 10.1074/jbc.M704629200. More molecular insights on the mechanisms through which 4E-T induced meiosis resumption will strengthen the impact and the novelty of this study (See two key experiments in Figure 1).

The authors want to thank the reviewer for this positive feedback on our manuscript. While we agree that an interaction of 4E-T with Zar1l (Heim et al., 2022, PMID: 36278895) and PATL2 (e.g. Minshall et al., 2007, PMID: 17942399 or Nakamura et al., 2010, PMID: 20471969) has been observed before, we believe that in the current manuscript we describe several important conceptual advances. Most importantly, although 4E-T has been described to be in a complex with Zar1l/PATL2 in *Xenopus* oocytes, the functional relevance of these complexes for the maintenance of the prophase-I arrest in full-grown oocytes was unknown. By performing for the first time 4E-T knock-down experiments in full-grown vertebrate oocytes after an unperturbed early oogenesis, we show that 4E-T is essential to maintain the prophase-I arrest. Depletion of Zar1l only accelerated meiotic resumption upon hormonal stimulation but did not induce spontaneous meiotic resumption (Heim et al., 2022, PMID: 36278895). Thus, in contrast to Zar1l, 4E-T is a key translational regulator essential for the prophase-I arrest of oocytes. While the excellent Minshall et al., 2007 publication from the Standard lab and its characterization of the CPEB1/4E-T/PATL2 complex impacted the initial stages of our project, we want to emphasize that they did not perform loss-of-function studies and therefore only limited conclusions on the physiological relevance of the CPEB1/4E-T/PATL2 complex can be drawn from this publication.

Furthermore, in the present manuscript we characterize interactions 4E-T forms with other proteins, e.g. eIF4E or PATL2, and analyze at the molecular level their relevance for target mRNA binding, protein complex formation and the prophase-I arrest.

Rational and design of the experiments:

The logic of the experiments is solid, the experimental plans is well designed and appropriate controls are used. However, the results presented in many panels look preliminary, since conclusions are often based on the qualitative observation of the single replicate shown, without quantification of the western blot signals and statistical analysis. Therefore, without a statistical analysis of the quantified signals, it is impossible to evaluate if all the conclusions of the authors are supported by the data (see figure by figure comments).

Major comments:

1. Figure 1. In this figure the authors show that 4E-T acute depletion induces meiosis resumption in *Xenopus* and mouse oocytes. However, the molecular mechanisms through which this happens is only explored indirectly. Indeed, the authors show that the depletion of 4E-T accelerates meiosis resumption induced by progesterone by advancing the Mos accumulation, and potentially Cyclin B1. However, this experiment does not tell whether the acute reduction of 4E-T is sufficient to produce the translation de-repression of endogenous Mos or if the translation activation of endogenous Mos requires also progesterone. Additionally, it does not tell if the slow spontaneous maturation is due to the precocious accumulation of either Mos or Cyclin B1 or other unknown proteins. Since the 4E-T depletion is the most novel result of this paper, I think it deserves a more detailed experimental investigation, through two key experiments.

First, authors should investigate if acute 4E-T depletion in prophase oocytes induces the accumulation of endogenous Mos and Cyclin B1 proteins in the absence of progesterone. Since, 4E-T depletion induces meiotic maturation, Cdk1 activation has to be prevented with RO-3306

(similarly to what is done in the accompanying paper for DDX6 acute depletion), or by expressing a Cdk1 inhibitor as p21 (10.1091/mbc.10.10.3279) or Wee (PMID: 10673504).

This is a great experiment, and we are therefore grateful for the reviewer's suggestion. At the time when we initially performed these experiments, we had no antibody available that was sufficiently sensitive to detect cyclin-B1 in *Xenopus* oocytes. Meanwhile, we raised and validated a new antibody against *Xenopus* cyclin-B1 (**new** Supplementary Fig. 6c). With this antibody in hand, we now analyzed and quantified the protein levels of cyclin-B1 and c-Mos upon depletion of 4E-T in oocytes treated with the Cdk1 inhibitor Roscovitine. We observed a very strong upregulation of cyclin-B1 protein and a more modest increase in c-Mos protein (**new** Fig. 2e). We assume that the levels of c-Mos only modestly increase because under these experimental conditions c-Mos cannot be phosphorylated by Cdk1/cyclin-B, which, as shown previously (Castro et al., 2001, PMID: 11553706), protects c-Mos from degradation. We changed the text accordingly to describe this important experiment in detail (line 185-194).

2. Second, authors should test if blocking the translation of Mos or CyclinB1 with antisense delays or blocks meiosis resumption induced by 4E-T. It was published that inhibiting the translation of both Mos and Cyclin B1 completely blocks progesterone-induced meiosis resumption (10.1038/sj.embor.7400611). It will be important to test if this is true when 4E-T is acutely depleted. This epistatic analysis is important to determine if 4E-T depletion induces meiosis resumption through the translation de-repression of Mos/Cyclin B1 without progesterone, or if it is through the translation de-repression of additional proteins, as for example Ringo/Speedy.

We thank the reviewer for suggesting this experiment. We performed the suggested experiment and used established DNA antisense oligonucleotides targeting all cyclin-B mRNAs (Hochegger et al., 2001, PMID: 11585805). WB analysis confirmed that these antisense oligonucleotides, but not the corresponding sense oligonucleotides, prevented cyclin-B1 expression upon 4E-T depletion (**new** Fig. S2a). Quantifications revealed that suppressing cyclin-B1 synthesis significantly reduced the rate of spontaneous, PG-independent meiotic resumption in 4E-T-depleted oocytes (**new** Fig. S2b). To analyze the requirement for c-Mos expression, we inhibited its only known downstream target MEK by the addition of the small molecule inhibitor U0126. Similar to the conditions where we suppressed cyclin-B1 synthesis, inhibition of MEK strongly suppressed spontaneous meiotic resumption in 4E-T-depleted oocytes (**new** Fig. S2c). Thus, similar to what was reported for meiotic resumption under physiological conditions, i.e., PG-induced, (Haccard et al., 2005, PMID: 16374506), we observed that both treatments significantly affected spontaneous meiotic resumption upon 4E-T depletion, but neither treatment alone was sufficient to completely block it (line 194-203). Unfortunately, due to technical challenges, i.e., high rates of oocyte deaths, we were unable to simultaneously apply TRIM-away, oligonucleotides and MEK inhibitor to analyze the ability of 4E-T-depleted oocytes to undergo meiotic maturation when cyclin-B1 synthesis and MEK activity are inhibited.

3. Fig. 1B: Statistical analysis is missing.

We appreciate this comment and have added a quantification of spontaneous meiotic resumption including statistical analysis (**new** Fig. 1b).

4. Fig. 1G: The representation of the meiotic maturation curves as mean and SD is confusing. This problem becomes more evident in figure 5B, where it is hard to visualize the data. The objective of these rescue experiments is to evaluate the time of meiosis resumption in the

presence or absence of 4E-T. It would be more appropriate to compare the times of 50% GVBD among the different conditions. Furthermore, this will allow to perform a statistical analysis of the results essential for drawing any conclusions.

We thank the reviewer for this helpful comment and modified the manuscript accordingly. We quantified the rate of oocytes undergoing GVBD and the median time oocytes need to undergo GVBD in these experiments including a statistical analysis (**new Fig. 1g**).

5. Fig. 1H: It would be important to evaluate the accumulation of endogenous Cyclin B1 in these oocytes. Since Cyclin B1 mRNA is also recruited to this inhibitory complex and Cyclin B1 overexpression is sufficient to induce meiotic resumption, it might be an increase in Cyclin B1 accumulation the reason of the spontaneous maturation.

We agree with the reviewer that this is an important point. As aforementioned (see point 1), we now have an antibody available, which we can use to detect cyclin-B1 in *Xenopus* oocytes. Using the identical samples shown in Fig. 1h, we immunoblotted for cyclin-B1 and can now demonstrate that 4E-T depletion causes a strong increase in cyclin-B1 levels (Fig. 1h). Rescue experiments with ectopic 4E-T strongly decrease cyclin-B1 upregulation, a result that is strongly supported by our statistical analyses of the experiments shown in the new Fig. 2e. Thus, as suggested by the reviewer, these experiments indicate that an increase in cyclin-B1 levels contributes to spontaneous meiotic resumption in the absence of 4E-T (line 132-134).

6. Fig. 1J: Similar problem than Fig. 1G.

We agree and accordingly quantified the rate of oocytes undergoing GVBD and the median time oocytes need to undergo GVBD in these experiments (**new Fig. 1j**).

7. Fig. 2A: Quantification of the puromycin signals on the different replicates of the experiment and statistical analysis are missing before concluding on the increase of global translation upon 4E-T acute depletion.

We agree and now included a quantification of the puromycin signal and performed a statistical analysis of the results (**new Fig. 2a**).

8. Fig. 2B: Interestingly, *wee2* mRNAs is included in the 4E-T inhibitory complex. It would be interesting to evaluate if *wee2* mRNA is translated upon 4E-T acute depletion. This would be important since *wee2* expression in prophase I could arrest/slow down meiosis resumption.

This is an interesting aspect, which we addressed by performing *Wee2* western blot of 4E-T-depleted oocytes treated with Roscovitine. Indeed, we observed a tendency for upregulation of *Wee2* (see below), although the result did not reach statistical significance. Interestingly, we noted that depletion of 4E-T slightly increased the level of inhibitory Cdk1 phosphorylation in oocytes that have not yet undergone GVBD (e.g. in **Fig. 1h and 2c**), which could be a consequence of upregulated *Wee2*. While we think that this is an interesting hypothesis, we believe that substantial further studies are required to corroborate this finding and dissect a potential molecular function of 4E-T in *Wee2* mRNA translational activation. We, therefore, decided not to include these novel data in the manuscript, but rather modify the text to refer and mention the enrichment of *Wee2* mRNA in the 4E-T IP samples and possible consequences from this in more detail (line 213-217).

9. Fig. 2D: The statistical analysis is missing. The labeling of the Y-axis is confusing.

We agree. We added a statistical analysis and relabeled the y-axis (new Fig. 2d).

10. Fig. 2E: It will be Important to show in the input lines, the abundance of Mos, Cyclin B1 and Cyclin A1 proteins by western blot. This will strengthen the observation reported showing that the timing of the protein accumulation matches the timing when their mRNAs are released from 4E-T. The statistical analysis of the mRNA quantifications is missing.

We thank the reviewer for this suggestion. Unfortunately, we used all samples to generate the data shown in Fig. 2e and are, therefore, not able to re-probe them for c-Mos, cyclin-B1 and cyclin-A1. However, for the following reasons we question if these western blots would provide valuable insights. As shown previously (Santoni et al., 2024, PMID: 38358892; Piqué et al, 2008, PMID: 18267074; Mendez et al., 2002, PMID: 11927567), accumulation of c-Mos and cyclin-B1 protein levels depends on a complex interplay of translation and stabilization. Specifically, Santoni et al. demonstrated that PG-triggered translational activation of c-Mos happens before Cdk1 activation/GVBD, which is in line with our results showing that 4E-T and c-Mos mRNA dissociated before Cdk1 activation (Fig. 2e; Fig. 2f in revised version). Yet, c-Mos protein accumulation occurs only after GVBD (Santoni et al., 2024, PMID: 38358892), probably due to the requirement of Cdk1/cyclin-B-mediated phosphorylation of c-Mos, which protects it against degradation (Castro et al., 2001, PMID: 11553706). In line with our data (Fig. 2e; Fig. 2f in revised version), Santoni *et al.* revealed that translational activation of cyclin-B1 happens after Cdk1 activation, while cyclin-B1 protein levels increase before GVBD. Thus, the timing of translational activation of c-Mos and cyclin B1 mRNAs does not match the timing of the accumulation of the respective proteins. We, therefore, do not expect that their levels in the input samples would provide useful information. This applies in particular for cyclin-A1, because 4E-T depletion did not significantly upregulate translation of a cyclin-A1 reporter mRNA (Fig. 2d). As discussed in the manuscript, we speculate

that translational activation of cyclin-A1 mRNA is controlled by additional, yet to be identified, regulatory mechanisms.

As suggested, we added the statistical analysis for Fig. 2e (new Fig. 2f).

11. Fig. 3A-B: Some of the interaction looks very variable in the two illustrated replicates: CPEB1 in Fig.3A looks almost completely RNA-dependent, but in Fig 3B: CPEB1 interaction with 4E-T is instead almost completely resistant to RNAse treatment at 0h.

We thank the reviewer for these precise observations and accordingly quantified and normalized all results from Fig. 3a and 3b and performed statistical analyses (**new** Fig. 3c and **new** Fig. S3a). Additionally, we performed a third replicate of Fig 3b. As pointed out by the reviewer, it was key to normalize the levels of the 4E-T interactors to the level of 4E-T immunoprecipitated. These quantifications revealed that the interaction of CPEB1 with 4E-T – although to a variable degree – is highly sensitive to RNaseA treatment in both Fig. 3a and 3b.

In Fig 3B, it looks odd that Zar1L is recovered at lower level at 0h in the absence of RNase treatment as compared to the RNase treated sample. This last observation could be due to the lower efficiency of the 4E-T IP in samples not treated with RNase. However, if the difference in the efficiency of 4E-T IP is taken in account, PATL2 would be higher at 0h in the sample not treated with RNase as compared to the one treated, contradicting the conclusion of the authors that PATL2 interaction is RNase independent. The western blot signals need to be quantified in the different replicates of these experiments. The levels of the interactors of 4E-T should be normalized to the level of 4E-T IP-ed. A statistical analysis is critical to determine which interactions are dependent or not on RNA, and which interactions are stable or not during meiotic resumption. Without this analysis, it is difficult to reach any conclusions.

As aforementioned, we repeated the experiment shown in Fig. 3b, normalized the data to the amount of immunoprecipitated 4E-T and performed statistical analyses of all IP experiments (**new** Fig. 3c and **new** Fig. S3a). These analyses revealed a tendency for an increased interaction with Zar1l upon RNaseA treatment, although this did not reach statistical significance due to considerable variation between experiments. We speculate that the loss of some interactions by RNaseA treatment could make 4E-T more accessible for other interactions not relying on RNA, e.g. with Zar1l. The quantifications also revealed that the interaction with PATL2 was in fact partially RNaseA sensitive, as suggested by the reviewer. Of note, this novel insight did not impact our working model (Fig. 6c) but could indicate that PATL2 binds to a given mRNA not only as part of a 4E-T complex, but also as part of an additional, 4E-T independent complex potentially fulfilling different mRNA regulatory functions. Alternatively, PATL2 could require multiple protein-RNA and protein-protein interactions to be incorporated into the complex with a sufficiently high affinity. To better represent the complex interaction network, we decided to omit the table in Fig. 3c but rather display the quantifications for each protein including statistical analyses (new Fig. 3c and new Fig. S3a).

12. Fig. 3B: It was reported that there are multiple eIF4E expressed in oocytes (10.1074/jbc.M704629200,10.1101/gad.350400.123.,10.1038/s44319-023-00006-4). Interestingly, different eIF4E proteins bind to distinct set of mRNAs to produce translation activation or inhibition (10.1038/s44319-023-00006-4). Is the antibody used in this study able to

distinguish these different eIF4Es? It would be important to evaluate which eIF4E isoform is included in this complex since they have dramatically opposite effects on translation.

We agree with the reviewer that this is an interesting point. However, we used a commercially available antibody (Cell Signaling #9742) that recognizes an antigen region (surrounding serine-209 of eIF4E), which is well conserved within the eIF4E protein family. Therefore, this antibody is probably not specific for any of the eIF4Es. Based on this fact, we use the general term “eIF4E” throughout the manuscript, although we are well aware that the individual isoforms might have an opposing effect on the translation state of the bound mRNA (e.g., Lorenzo-Orts et al., 2024, PMID: 38177902).

13. Fig. 3C: This table is important to well summarize the data from the previous two experiments. However, because of the qualitative approach used in Fig. 3A-B, at the current stage, this summary is not completely supported by the data.

We fully agree with the reviewer. Since we performed an additional biological replicate for Fig. 3b and extensive statistical analyses (see point 11), we came to the conclusion that a simple table does not satisfy the complex pattern of interactions. Therefore, we removed this table and replaced it by the individual quantifications for each protein (**new** Fig. 3c).

14. Fig. 4B-C-D: The western blot signals need to be quantified in the different replicates of these experiments. The levels of the interactors of 4E-T should be normalized to the level of 4E-T IPed. A statistical analysis is critical to determine which interactions are taking place or not upon deletion of the different domains. Without this analysis, it is hard to drive conclusions.

We agree with the reviewer that these are important points. To adequately address them, we performed one additional replicate of Figs. 4b and 4c, quantified all data from Fig. 4b-d and included a statistical analysis (**new** Fig. 4e and **new** Fig. S4a).

15. Some of the contradictions between the replicates shown are:

In Fig. 4B, PATL2 and LSM14A binding is strongly decreased in 4Ec-mutant but almost unchanged in 4D. Zar2 binding to 4Ec-mutant mutant seems unchanged in Fig. 4C but completely lost in Fig. 4D.

As pointed out by the reviewer rigorous quantifications including data normalization are required to correctly interpret the data. To address this, we performed an additional biological replicate and normalized the signal intensities to the one of the immunoprecipitated Flag-4E-T variants. These analyses revealed that the interactions between 4E-T and PATL2 or LSM14A were not significantly reduced in the 4ET Δ 4Ec-mutant (**new** Fig. 4e and **new** Fig. S4a), although there was some variation between the individual replicates in both Fig. 4b and 4d. The interaction between Zar2 and 4ET Δ 4Ec-mutant was significantly reduced compared to 4E-T WT in both Fig. 4c and 4d, although the extent of this effect was indeed slightly bigger in Fig. 4d (for quantifications see the **new** Fig. 4e and **new** Fig. S4a).

16. Fig. 5B is very confusing. Please see comments on Fig. 1G. Without a statistical analysis of the time of 50% GVBD, it is hard to assess if the conclusions of the authors are supported by the data. The plotting of the time of 50% GVBD for all conditions will also avoid repeating the control curves multiple times and reduce the complexity of the figure legend.

We thank the reviewer for this comment, which was also raised by reviewer #1. We, therefore, completely reformatted this figures by now showing the percentage of oocytes undergoing meiotic resumption and the time they need to undergo GVBD including statistical analyses (**new** Figs. 5b and 5d). As suggested by reviewer #1, we now focus on the relevant 4E-T mutants identified in Fig. 4 ($\Delta 4Ec$, $\Delta 4Enc$, $\Delta PATL$ and $\Delta LSM14A$). This novel representation nicely illustrates that the $\Delta LSM14A$ mutation fully rescued meiotic timing, while the other mutants show a tendency for incomplete rescue, i.e. they are not statistically different from either the 4E-T depletion or the WT rescue condition (for clarity, we only show p-values comparing the different mutant rescue conditions with the WT rescue). This provides a rational for creating and testing the $\Delta 4Ec \Delta PATL$ double mutant of 4E-T, which is indeed completely inefficient in rescuing depletion of 4E-T (Figs. 5c, 5d and S5a).

17. In Fig. 5F, the statistical analysis is missing to evaluate if the author conclusions are supported by the data.

We added a statistical analysis to this figure, which strongly supports our conclusions (new Fig. 5f).

18. In fig. 6B, the statistical analysis is missing to evaluate if the author conclusions are supported by the data.

We added a statistical analysis to this figure (new Fig. 6b), which fully corroborate our conclusions.

**Reviewer #1 (Remarks to the Author):**

Although the authors have revised the manuscript according to the comments, the major concerns have not been solved.

1. The authors have verified the role of 4E-T in meiotic arrest in mice, however their observations seem to contradict previous findings: haploid loss of 4E-T leading to a decrease in GVBD rate (PMID: 38088064); knockdown of 4E-T by siRNA resulting in the inability to undergo GVBD (PMID: 26147080).

We thank the reviewer for her/his insightful comment. The reviewer is correct in that PMID: 38088064 and 26147080 report that reduced levels of 4E-T lead to problems with GVBD. At a first glance, these observations appear to contradict our findings that loss of 4E-T promotes GVBD.

Of note, Melina Schuh, the corresponding author of PMID: 26147080, is also a co-author of our manuscript, and we are, therefore, fully aware of the published results. Melina Schuh and her team used an RNAi-approach **“to block protein expression (e.g., of 4E-T) from an early stage of oocyte development onwards”**. As mentioned by the reviewer, PMID: 38088064 used haploinsufficient mouse oocytes to study the consequences of partial 4E-T deficiency. Thus, both studies have in common that they drew conclusions based on oocytes that had experienced partial 4E-T loss throughout nearly the entire process of oogenesis. In fact, the authors of PMID: 38088064 state that the significantly altered transcriptome and translome they observed in 4E-T (aka Eif4enif1) haploinsufficient GV oocytes could be **“due to the cumulative effects of Eif4enif1 haploinsufficiency throughout oocyte development”**. We now discuss this important point in the discussion.

Given this, it is not surprising that these studies, though valid, reached conclusions that differ from our study, where we specifically depleted 4E-T from a distinct developmental stage, i.e., full grown prophase-I arrested oocytes, without touching its potential function during early oogenesis. Key for our discovery was the advent of TRIM-Away, which was established by Melina Schuh to acutely deplete endogenous proteins in living systems (PMID: 29153837).

Furthermore, if the function of 4E-T is conserved in vertebrates, more experiments in mice are required for validation, especially regarding the repetition of important mechanisms (such as 4E-T's regulation of Mos and Cyclin-B1 mRNA).

We appreciate the reviewer's comment. As shown in figure 1c-e, mouse oocytes lacking 4E-T underwent spontaneous meiotic resumption. Thus, the essential function of 4E-T for maintaining the prophase-I arrest in full grown oocytes is conserved in frog and mouse. To dissect the underlying mechanism, we focused on *Xenopus* rather than mouse oocytes because they are amenable to biochemical cell cycle analyses. Unfortunately, the author conducting the experiments in mouse oocytes (S. Cheng) left Melina Schuh's lab during the revision process to establish his own lab in China and we were therefore unable to study the mechanism in mouse oocytes in more detail. Thus, we agree with the reviewer that we did not formally demonstrate that 4E-T acts also as translational repressor in mouse oocytes. Of note, **in the accompanying manuscript** (Cheng and Schuh, 2024), the Schuh lab demonstrates that depletion of DDX6, which is part of the 4E-T complex, in mouse oocytes affects translation of cyclin-B1 in full agreement with our results in *Xenopus* oocytes.

Nevertheless, to take the reviewer's concern into account, we suggest changing the title from:

“Translational repression by 4E-T is crucial to maintain the prophase-I arrest in vertebrate oocytes”

to

“The translational repressor 4E-T is crucial to maintain the prophase-I arrest in vertebrate oocytes”.

This statement – not implying that the downstream mechanism of translational repression of cyclin-B1 is identical in mouse and frog oocytes – is in full agreement with the data shown throughout our manuscript. If the reviewer is bothered by the fact that we investigated 4E-T in only two vertebrate species, we would be willing to change “...in vertebrate oocytes” to “...in mouse and frog oocytes”.

2. Although the authors have performed many experiments on 4E-T, however the major findings have been reported previously.

We appreciate the reviewer’s comment. 4E-T has indeed been studied previously, but our findings go substantially beyond what was previously known. This study is the first to report the essential role of the translational regulator 4E-T in maintaining prophase-I arrest in fully grown oocytes, and we demonstrate that this fundamental function of 4E-T is conserved in frog and mouse oocytes. In contrast, none of the publications cited by the reviewer investigated the consequences of 4E-T or PATL2 loss-of-function. Instead, these studies share a common experimental approach—overexpression of ectopic 4E-T/PATL2—where any translational inhibitor, regardless of its physiological role, would lead to translational repression. Therefore, we disagree with the assertion that these studies have already established the necessity of 4E-T for the integrity of fully grown oocytes.

To support our argument, we respond in detail to each of the reviewer’s conclusions drawn from the mentioned papers.

For example, CPEB1/4E-T/PATL2 complex and its inhibitory effect on translation and involvement in meiotic recovery have been previously reported by Minshall et al., 2007 (PMID: 17942399).

The reviewer is correct that Minshall et al. demonstrated the presence of 4E-T in a complex with PATL2 and CPEB1 in frog oocytes. While this is an excellent study, it did not address the key question: Is the translation-repressing function of endogenous 4E-T essential for maintaining prophase-I arrest in oocytes?

Instead, the authors employed a highly artificial tethering assay to show that ectopic 4E-T can suppress the translation of an ectopic mRNA (Fig. 7). Since any unrelated ectopic translation repressor could similarly suppress the translation of a tethered, ectopic mRNA in oocytes, these experiments do not provide insight into the physiological function of 4E-T in oocytes. Furthermore, Minshall et al. did not conduct the critical experiments necessary to support the reviewer’s conclusion—namely, interfering with the function of CPEB1, 4E-T, or PATL2 to analyze phenotypic effects. Minshall et al. report only a slight impact on the timing of oocyte maturation following injection of α -eIF4E1b antibodies, and even then, only at a particular PG concentration. We want to emphasize that this is very different from our finding that loss of 4E-T causes ***PG-independent meiotic maturation***. For these reasons, we think that our findings are novel and provide important new insights into the translational regulation in oocytes highly relevant for researchers working on the regulation of the cell cycle and translation.

Furthermore, the role of PATL2 as an RNA-binding component of mRNP (including 4E-T) in meiotic maturation has been documented: regulating the translation of c-Mos and Cyclin B1 mRNA, similar to the mechanisms in this study (PMID: 20471969, 20826699).

We agree with the reviewer that PMID: 20471969 demonstrated that large protein complexes containing PATL2 bind c-Mos and cyclin-B1 mRNAs. However, other conclusions drawn by the reviewer are not supported by the data published in the cited papers.

- While both papers show that PATL2 can bind to RNA in general, neither PMID: 20471969 nor PMID: 20826699 investigated if PATL2, rather than any of the other known RNA-binding proteins in the complex, is indeed the critical c-Mos and cyclin-B1 mRNA-binding component of the 4E-T complex.
- PMID: 20471969 showed that ectopic PATL2 suppressed translation of ectopic mRNAs. However, all tested mRNAs *were completely unrelated to endogenous c-Mos or cyclin-B1 mRNAs* and the *translational state of endogenous c-Mos or cyclin-B1 mRNAs* (or corresponding reporter mRNAs containing their UTRs) *has not been studied*. Therefore, the conclusion that PATL2 regulates the translation of c-Mos and cyclin-B1 is not supported by the published data.
- PMID: 20471969 showed that overexpression of PATL2 delayed PG-induced meiotic resumption as well as the expression of cyclin-B1 and c-Mos. However, since Cdk1 activity stimulates the translational activation of both c-Mos and cyclin-B1 (PMID: 38358892), any factor delaying meiotic resumption and Cdk1 activation would naturally impair their translation. Thus, to thoroughly demonstrate that PATL2 directly regulates c-Mos and cyclin-B1 translation it would be necessary to deplete PATL2 from oocytes kept in prophase-I arrest by a Cdk1 inhibitor.
- PMID: 20826699 used an artificial tethering assay to show that ectopic PATL2 suppressed translation of an ectopic reporter mRNA. As explained above, we believe such experiments do not allow conclusions about the *physiological function of PATL2 in regulating c-Mos and cyclin-B1 in oocytes*.

Moreover, the regulation of c-Mos mRNA translation by PATL2 has also been reported in mice (PMID: 33614659).

Although the abstract of PMID: 33614659 explicitly states that "...PATL2 bound mRNAs of Mos...", the authors **do not conduct any experiments** to demonstrate that PATL2 actually binds to c-Mos mRNA in mouse oocytes. The main focus of the publication is to understand the mechanism underlying female infertility associated with a mutation in PATL2 (Y217N). The authors observe that overexpression of mutant PATL2^{Y217N}, but not wildtype PATL2, leads to lower c-Mos protein levels (see below Fig.5A) and suppresses translation of a c-Mos reporter mRNA in oocytes. Without showing the levels of ectopic PATL2 in these experiments, the authors claim that this effect is due to higher protein levels of PATL2^{Y217N} compared to WT PATL2. However, even if PATL2^{WT} is not as strongly overexpressed as mutant PATL2 under these experimental conditions, we are surprised that c-Mos expression is not reduced at all compared to the negative control not expressing ectopic PATL2 (see below). Thus, we are puzzled about the reviewers comment that regulation of c-Mos translation by *endogenous wild-type* PATL2 has been shown in this publication. In fact, the authors in this publication speculate that "As MOS does not seem to be essential for the initiation of oocyte maturation, it suggests that **the oocyte maturation arrest observed in patient with PATL2^{Y217N} mutation may be caused by some other mechanisms.**" We, therefore, disagree with the claim that PMID: 33614659 provided convincing data demonstrating that PATL2 regulates c-Mos translation.

Fig. 5A from PMID: 33614659 showing c-Mos and actin immunoblot analyses of mouse oocytes injected with cRNA encoding WT or Y217N PATL2 and negative control oocytes.

3. Proteomic analysis of 4E-T knockdown oocytes and RIP experiments for PATL2 may give new insights into the molecular mechanisms and probably promote the novelty of the manuscript, however the authors have failure to address these comments.

We appreciate the reviewer's comment. To address this, we performed MS analyses on lysates from Ctrl- or 4E-T-depleted prophase-I arrested oocytes to identify additional proteins under the control of 4E-T. As shown in **new Fig.S3**, we identified two clusters of proteins (cluster 1 and 2) that were strongly upregulated upon 4E-T depletion. KEGG pathway analysis revealed that these clusters are highly enriched in pathways governing "Cell cycle," "Oocyte meiosis," and "Progesterone-mediated oocyte maturation," reinforcing our hypothesis that 4E-T plays a crucial role in suppressing the translation of mRNAs driving meiotic resumption. Additionally, we observed a significant upregulation of other key molecular pathways, including "mRNA surveillance," "Spliceosome," "DNA replication," "Nucleocytoplasmic transport," and "Endocytosis," suggesting that 4E-T also regulates these processes during meiotic maturation and early embryonic divisions.

Further in-depth analysis of clusters 1 and 2 revealed a strong upregulation of factors involved in MT network regulation, motor proteins, and histone proteins following 4E-T depletion. Given their essential role in proper cell division, this further supports the idea that 4E-T negatively regulates mRNAs critical for meiotic and early mitotic divisions. While the sensitivity of our measurements was insufficient to unambiguously detect peptides corresponding to cyclin-B1 or c-Mos, our new MS data provide a strong foundation for identifying additional candidate mRNAs and biological processes under 4E-T control. A deeper investigation into these findings is beyond the scope of this project and will be pursued in future studies.

4. The authors explained that incomplete restoration of translation in Fig. 2a is due to depletion of proteins interacting with 4E-T. Thus, protein expression in the rescue group should be added to Fig. S1c.

We appreciate the reviewer's comment. To address this point, we analyzed under depletion and rescue conditions the levels of Zar1l, which was the most significantly do-depleted interactor (Fig. S1c). Quantification of three biological replicates revealed that re-expression of TRIM-resistant 4E-T did not restore the levels of Zar1l to the ones in control depleted oocytes (**new figure Fig. S1d**). Since Zar1l has been reported to function as a negative regulator of translation and its loss weakens the prophase-I arrest (PMID: 36278895), reduced levels of Zar1l under rescue conditions provides a molecular explanation for the imperfect rescue observed in figure 2a.

New Figure S1D: Xenopus stage-VI oocytes were injected with water or mRNA encoding Flag-4E-T^{TRIM}. 18h after injection, oocytes were co-injected with mRNA encoding Flag-TRIM21 and either 4E-T^{Ab1} or unspecific control (Ctrl) antibodies. 22h after the second injection, oocytes were lysed for immunoblotting as indicated. **Left:** One representative experiment of three independent biological replicates is shown. **Right:** Zar1l signals were quantified and normalized intensities from three independent biological replicates are given as mean±s.d.. p-values were calculated using one-way ANOVA with Tukey's multiple comparisons test.

Reviewer #2 (Remarks to the Author):

I appreciate the work done by the authors to modify the figures and to perform the two additional experiments that strengthen the logic of this manuscript. The revised manuscript displays now quantifications and a statistical analysis of the experiments. I think these results would be interesting for the readership of Nature Communications.

We thank the reviewer for his/her constructive feedback and the positive assessment of our study.

However, two last concerns remain:

1) Authors used throughout the paper a series of pairwise t-test to compare multiple conditions of the same experiment, which is not appropriate. Indeed, the nature communication guidelines suggest: “Multiple comparisons: When making multiple statistical comparisons on a single data set, authors should explain how they adjusted the alpha level to avoid an inflated Type I error rate, or they should select statistical tests appropriate for multiple groups (such as ANOVA rather than a series of t-tests).” An ANOVA test with multiple comparisons is advised to assess if the conclusions of the authors are statistically supported.

We thank the reviewer for this comment and accordingly provide *novel statistical analyses* with ANOVA or t-tests with multiple comparisons in the second revised version of our manuscript. These novel statistical analyses (*new* Figs. 1-6 and Figs. S1, S2, S4-S6) are in full agreement with our initial assessment of the results presented throughout our manuscript.

2) In figure 5b, the authors are displaying the comparison between the ability to rescue of the WT and the different mutants. Authors should instead compare the median time of the rescue conditions (WT or the mutants) with the median time of the TRIM-Away condition, which is the point of the rescue.

We appreciate the reviewer’s comment. In the revised version of our manuscript, we now compare the median time of the rescue (WT/mutant) conditions with the median time of the TRIM-Away condition (see *new* Fig. 5b).

REVIEWER COMMENTS:

First, we want to thank both reviewers for their positive and constructive evaluation of our revised manuscript. Their insightful comments and expert suggestions greatly improved this study.

Reviewer #1 (Remarks to the Author):

The authors have addressed my major concerns in the revisions.

We thank the reviewer for the positive evaluation of our manuscript and for taking the time to help us improve our manuscript—we really appreciate the support and helpful suggestions

Reviewer #2 (Remarks to the Author):

I thank the authors for having corrected the statistical analyses throughout the paper. However, two important concerns remain, particularly regarding the mass spectrometry (MS) experiment the authors performed during the revision to evaluate differences between the proteomes of control and 4E-T–trimmed oocytes and Fig. 5.

Thank you to the reviewer for taking the time to evaluate the new experiments and for the helpful input that contributed to further improving our manuscript. As detailed below we have now reanalyzed the MS experiment according to the suggestions.

a) Concerns regarding the MS experiment and data analysis:

1. Data analysis: The use of Z-score normalization is suitable for pattern recognition (e.g., clustering of time resolved samples), but not for assessing differential expression between two conditions on three biological replicates. I recommend that the authors perform a statistical comparison by calculating log₂ fold changes and applying a significance test with FDR correction, presenting the results as a volcano plot or an equivalent. This will allow proper evaluation of the magnitude and statistical significance of observed differences.

We thank the reviewer for this comment. As suggested, we have reanalyzed our MS results including a statistical analysis (**new Fig. S3b**), which facilitates evaluation of the results for individual proteins. As detailed in the text of our revised manuscript, we still find several histone proteins as well as regulators of the microtubule spindle and DNA replication among the significantly enriched proteins upon 4E-T depletion. These proteins have a central role during meiotic maturation and the following early embryonic divisions and we are therefore confident that studying their dependence on 4E-T will be the starting point for exciting studies in the future.

2. Availability of raw data: More raw MS counts should be provided as supplementary figures. Additionally, the raw data are still not available on PRIDE, as required.

We apologize for the confusion. Following the guidelines of *Nature Communications*, we had uploaded the MS raw data on PRIDE in the previous round of revision. As stated in the data availability section, these data are available to the reviewers upon login on the PRIDE website using a specific reviewer account:

Username: reviewer_pxd062261@ebi.ac.uk

Password: EH8UyUZfHsV

Upon acceptance of the manuscript, the data will be publicly accessible without the need to login.

3. Upon looking at the available data, several RNA-binding proteins (RBPs), including *lsm14b.L*, *zar1.L*, *zar1.S*, *zar1l.L*, and *patl2.L*, appear less abundant in the TRIM condition. These changes could potentially contribute to the observed deregulation of translation and should be discussed if they are statistically significant. They could suggest a general destabilization of the complex that includes 4ET upon its removal.

We thank the reviewer for this comment, which had also been raised by reviewer #1 in the previous round of revision. As shown in Fig. S1c, interactors of 4E-T were found to be destabilized to various extents by 4E-T depletion, with *Zar1l* and *Zar2* being the only ones to be significantly reduced in their levels. Such a co-depletion of interactors in multi-protein complexes is not an unusual phenomenon (Juszkiewicz and Hegde, 2018, *Mol Cell*, PMID: 30075143). As shown in Fig. S1d, reduced levels of *Zar1l* could not be rescued by expression of ectopic 4E-T. As suggested by the reviewer, we discuss that this may explain our inability to fully rescue the 4E-T depletion phenotype under certain experimental conditions, such as the increased translational activity observed following 4E-T depletion (Fig. 2a, and lines 157-165).

4. Specificity of TRIM-away: The clustering analysis shows that many ribosomal proteins are grouped into clusters 6 and 5 (the same clusters as 4E-T, which the authors have confirmed to be strongly depleted by TRIM-away). This is concerning and might indicate that the TRIM-away approach is depleting abundant proteins non-specifically, potentially affecting the general conclusions of the study. Without statistical validation (as requested in point 1), it is not possible to assess whether the differences detected are real or experimental noise. If statistical analysis confirms that some ribosomal proteins are significantly depleted, this must be discussed explicitly in the paper.

We thank the reviewer for this careful analysis of our initial MS results. Following the reviewer's suggestion, we have reanalyzed the MS data including a statistical analysis. This revealed that no ribosomal subunit was among the proteins significantly decreased upon 4E-T depletion (see Source Data Excel File Figure S3b). This finding aligns with our observation that 4E-T loss led to

increased translational activity (e.g. Fig. 2a), contrary to what would be expected if ribosomal protein levels were significantly reduced.

5. Validation of MS findings:

As also stated by the authors, most proteins/mRNAs experimentally studied by the authors as deregulated by 4E-T TRIM are not detected in the MS dataset, as CcnB1 and Mos. The authors should validate at least some of the proteomic changes by western blot. Interestingly, TPX2 appears more abundant in TRIM samples; if this is statistically significant, it would support the 4E-T IP/qPCR findings presented. Overall, without proper statistical analysis and independent validation (as by western blot), the following statements are not currently justified:

- “Mass spectrometry (MS) analyses identified additional proteins with diverse cellular functions to be upregulated in 4E-T depleted oocytes.” Lines 218–241.

We thank the reviewer for this insightful suggestion and agree that TPX2 would be a good candidate to validate our MS results by western blot. Unfortunately, we currently have no antibody available to reliably detect endogenous *Xenopus* TPX2 in oocyte samples. To circumvent this limitation, we created a reporter mRNA using the 3'UTR of TPX2. Importantly, translation of this reporter mRNA was significantly upregulated upon depletion of 4E-T, and this could be fully rescued by expression of ectopic 4E-T (**new Figs. S3c and S3d**). Furthermore, in the revised version of our manuscript we restricted description of individual proteins to the ones significantly upregulated in the new analysis.

b) Concerns regarding Figure 5b:

In Figure 5b, the correct statistical analysis shows no significant difference between the TRIM-away condition and its rescue with various 4E-T fragments (p-values ranging from 0.28 to 0.699), except for the LSM14 mutant ($p < 0.05$). Therefore, the experiment must be repeated with more replicates, or the authors should revise their conclusion. For example, the statement: “Compared to the non-rescue condition, all tested single-deletion mutants were able to rescue meiotic timing to a certain extent with the Δ LSM14A variant being as efficient as WT 4E-T (Figs. 5a and 5b)” is not supported by the current data.

We apologize for the confusion. The aim of this experiment was to find 4E-T mutants that could not rescue the meiotic timing phenotype of 4E-T-depleted oocytes. To achieve this, we followed the suggestion raised in the previous round of revision and compared the meiotic timing of 4E-T-depleted oocytes expressing the different 4E-T rescue constructs to the one of 4E-T-depleted oocytes injected with the water control (Fig. 5b). These analyses revealed that none of the single mutants – except for the Δ LSM14A variant – gave rise to a meiotic timing that was statistically significantly different from the one of the water control, i.e. they were all inefficient in rescuing the meiotic timing phenotype. Yet, while not being statistically significant, we observed a trend

towards longer GVBD timings that approached the timing observed in WT-expressing oocytes (Fig. 5b) suggesting that the mutants might not be fully inactive. This led to the creation of the $\Delta 4E-T \Delta PATL$ double mutant used in Figs. 5c and 5d, which was consistently unable to rescue 4E-T depletion. To prevent any misunderstandings, we rephrased this part to *“Compared to the non-rescue condition, the single-deletion mutants, except for the $\Delta LSM14A$ variant, were not able to significantly rescue timing of GVBD (Figs. 5a and 5b). Nevertheless, all tested constructs still showed a trend towards prolonged duration of meiotic resumption. We, thus, combined the $\Delta 4E-T$ and $\Delta PATL$ deletions and analyzed the ability of the double deletion mutant to rescue meiotic timing in 4E-T-depleted oocytes. Notably, the $\Delta 4E-T/\Delta PATL$ double mutant did not rescue at all (Figs. 5c, 5d and S6a).”*